# METANORM: LEARNING TO NORMALIZE FEW-SHOT BATCHES ACROSS DOMAINS

**Yingjun Du[1], Xiantong Zhen[1,2], Ling Shao[2], Cees G. M. Snoek[1]**
[1]AIM Lab, University of Amsterdam
[2]Inception Institute of Artificial Intelligence

## ABSTRACT

Batch normalization plays a crucial role when training deep neural networks. However, batch statistics become unstable with small batch sizes and are unreliable in the presence of distribution shifts. We propose MetaNorm, a simple yet effective meta-learning normalization. It tackles the aforementioned issues in a unified way by leveraging the meta-learning setting and learns to infer adaptive statistics for batch normalization. MetaNorm is generic, flexible and model-agnostic, making it a simple plug-and-play module that is seamlessly embedded into existing meta-learning approaches. It can be efficiently implemented by lightweight hypernetworks with low computational cost. We verify its effectiveness by extensive evaluation on representative tasks suffering from the small batch and domain shift problems: few-shot learning and domain generalization. We further introduce an even more challenging setting: few-shot domain generalization. Results demonstrate that MetaNorm consistently achieves better, or at least competitive, accuracy compared to existing batch normalization methods.

## 1 INTRODUCTION

Batch normalization (Ioffe & Szegedy, 2015) is crucial for training neural networks, and with its variants, e.g., layer normalization (Ba et al., 2016), group normalization (Wu & He, 2018) and instance normalization (Ulyanov et al., 2016), has thus become an essential part of the deep learning toolkit (Bjorck et al., 2018; Luo et al., 2018a; Yang et al., 2019; Jia et al., 2019; Luo et al., 2018b; Summers & Dinneen, 2020). Batch normalization helps stabilize the distribution of internal activations when a model is being trained. Given a mini-batch $\mathcal{B}$, the normalization is conducted along each individual feature channel for 2D convolutional neural networks. During training, the batch normalization moments are calculated as follows:

$$\mu_{\mathcal{B}} = \frac{1}{M}\sum_{i=1}^{M} a_i, \quad \sigma_{\mathcal{B}}^2 = \frac{1}{M}\sum_{i=1}^{M}(a_i - \mu_{\mathcal{B}})^2, \tag{1}$$

where $a_i$ indicates the $i$-th element of the $M$ activations in the batch, $M = |\mathcal{B}| \times H \times W$, in which $H$ and $W$ are the height and width of the feature map in each channel. We can now apply the normalization statistics to each activation:

$$a_i' \leftarrow \mathbf{BN}(a_i) \equiv \gamma \hat{a}_i + \beta, \quad \text{where,} \quad \hat{a}_i = \frac{a_i - \mu_{\mathcal{B}}}{\sqrt{\sigma_{\mathcal{B}}^2 + \epsilon}}, \tag{2}$$

where $\gamma$ and $\beta$ are parameters learned during training, $\epsilon$ is a small scalar to prevent division by $0$, and operations between vectors are element-wise. At test time, the standard practice is to normalize activations using the moving average over mini-batch means $\mu_{\mathcal{B}}$ and variance $\sigma_{\mathcal{B}}^2$. Batch normalization is based on an implicit assumption that the samples in the dataset are independent and identically distributed. However, this assumption does not hold in challenging settings like few-shot learning and domain generalization. In this paper, we strive for batch normalization when batches are of small size *and* suffer from distributions shifts between source and target domains.

Batch normalization for few-shot learning and domain generalization problems have so far been considered separately, predominantly in a meta-learning setting. For few-shot meta-learning (Finn

et al., 2017; Gordon et al., 2019), most existing methods rely critically on transductive batch normalization, except those based on prototypes (Snell et al., 2017; Allen et al., 2019; Zhen et al., 2020a). However, the nature of transductive learning restricts its application due to the requirement to sample from the test set. To address this issue, Bronskill et al. (2020) proposes TaskNorm, which leverages other statistics from both layer and instance normalization. As a non-transductive normalization approach, it achieves impressive performance and outperforms conventional batch normalization (Ioffe & Szegedy, 2015). However, its performance is not always performing better than transductive batch normalization. Meanwhile, domain generalization (Muandet et al., 2013; Balaji et al., 2018; Li et al., 2017a;b) suffers from distribution shifts from training to test, which makes it problematic to directly apply statistics calculated from a seen domain to test data from unseen domains (Wang et al., 2019; Seo et al., 2019). Recent works deal with this problem by learning a domain specific normalization (Chang et al., 2019; Seo et al., 2019) or a transferable normalization in place of existing normalization techniques (Wang et al., 2019). We address the batch normalization challenges for few-shot classification and domain generalization in a unified way by learning a new batch normalization under the meta-learning setting.

We propose MetaNorm, a simple but effective meta-learning normalization. We leverage the meta-learning setting and learn to infer normalization statistics from data, instead of applying direct calculations or blending various normalization statistics. MetaNorm is a general batch normalization approach, which is model-agnostic and serves as a plug-and-play module that can be seamlessly embedded into existing meta-learning approaches. We demonstrate its effectiveness for few-shot classification and domain generalization, where it *learns* task-specific statistics from limited data samples in the support set for each few-shot task; and it can also learn to generate domain-specific statistics from the seen source domains for unseen target domains. We verify the effectiveness of MetaNorm by extensive evaluation on few-shot classification and domain generalization tasks. For few-shot classification, we experiment with representative gradient, metric and model-based meta-learning approaches on fourteen benchmark datasets. For domain generalization, we evaluate the model on three widely-used benchmarks for cross-domain visual object classification. Last but not least, we introduce the challenging new task of *few-shot domain generalization*, which combines the challenges of both few-shot learning and domain generalization. The experimental results demonstrate the benefit of MetaNorm compared to existing batch normalizations.

## 2 RELATED WORKS

**Transductive Batch Normalization** For conventional batch normalization under supervised settings, i.i.d. assumptions about the data distribution imply that estimating moments from the training set will provide appropriate normalization statistics for test data. However, in the meta-learning scenario data points are only assumed to be i.i.d. within a specific task. Therefore, it is critical to select the moments when batch normalization is applied to support and query set data points during meta training and meta testing. Hence, in the recent meta-learning literature the running moments are no longer used for normalization at meta-test time, but instead replaced with support/query set statistics. These statistics are used for normalization, both at meta-train *and* meta-test time. This approach is referred to as *transductive batch normalization* (TBN) (Bronskill et al., 2020). Competitive meta-learning methods (e.g., Gordon et al., 2019; Finn et al., 2017; Zhen et al., 2020b) rely on TBN to achieve state-of-the-art performance. However, there are two critical problems with TBN. First, TBN is sensitive to the distribution over the query set used during meta-training, and as such is less generally applicable than non-transductive learning. Second, TBN uses extra information for multiple test samples, compared to non-transductive batch normalization at prediction time, which could be problematic as we are not guaranteed to have a set of test samples available during training in practical applications. In contrast, MetaNorm is a non-transductive normalization. It generates statistics from the support set only, without relying on query samples, making it more practical.

**Meta Batch Normalization** To address the problem of transductive batch normalization and improve conventional batch normalization, meta-batch normalization (MetaBN) was introduced (Triantafillou et al., 2020; Bronskill et al., 2020). In MetaBN, the support set alone is used to compute the normalization statistics for *both* the support and query sets at both meta-training and meta-test time. MetaBN is non-transductive since the normalization of a test input does not depend on other test inputs in the query set. However, Bronskill et al. (2020) observe that MetaBN performs less well for small-sized support sets. This leads to high variance in moment estimates, which is similar to the

difficulty of using batch normalization with small-batch training (Wu & He, 2018). To address this issue, Bronskill et al. (2020) proposed TaskNorm, which learns to combine statistics from both layer normalization and instance normalization, with a lending parameter to be learned at meta-train time. As a non-transductive normalization, TaskNorm achieves impressive performance, outperforming conventional batch normalization. However, it can not always perform better than transductive batch normalization. TaskNorm indicates non-transductive batch normalization estimates proper normalization statistics by involving learning in the normalization process. We also propose to learn batch normalization within the meta-learning framework, but instead of employing a learnable combination of existing normalization statistics, we directly learn to infer statistics from data. At meta-train time, the model learns to acquire the ability to generate statistics only from the support set and at meta-test time we directly apply the model to infer statistics for new tasks.

**Batch Normalization for Domain Adaptation and Domain Generalization** Domain adaption suffers from a distribution shift between source and target domains, which makes it sub-optimal to directly apply batch normalization (Bilen & Vedaldi, 2017). Li et al. (2016) proposed adaptive batch normalization to increase the generalization ability of a deep neural network. By modulating the statistical information of all batch normalization layers in the neural network, it achieves deep adaptation effects for domain-adaptive tasks. Nado et al. (2020) noted the possibility of accessing small unlabeled batches of the shifted data just before prediction time. To improve model accuracy and calibration under covariate shift, they proposed prediction-time batch normalization. Since the activation statistics obtained during training do not reflect statistics of the test distribution, when testing in an out-of-distribution environment, Schneider et al. (2020) proposed estimating the batch statistics on the corrupted images. Kaku et al. (2020) demonstrated that standard non-adaptive feature normalization fails to correctly normalize the features of convolutional neural networks on held-out data where extraneous variables take values not seen during training. Learning domain-specific batch normalization has been explored (Chang et al., 2019; Wang et al., 2019). Wang et al. (2019) introduced transferable normalization, TransNorm, which normalizes the feature representations from source and target domain separately using domain-specific statistics. Along a similar vein, Chang et al. (2019) proposed a domain-specific batch normalization layer, which consists of two branches, each in charge of a single domain exclusively. The hope is that, through the normalization, the feature representation will become domain invariant. Nevertheless, these normalization methods are specifically designed for domain adaptation tasks, where data from target domains are available, though often unlabelled. This makes them inapplicable to domain generalization tasks where data from target domains are inaccessible at training time. Seo et al. (2019) proposed learning to optimize domain specific normalization for domain generalization tasks. Under the meta-learning settings, a mixture of different normalization techniques is optimized for each domain, where the mixture weights are learned specifically for different domains. Instead of combining different normalization statistics, MetaNorm learns from data to generate adaptive statistics specific to each domain. Moreover, we introduce an even more challenging setting, i.e., few-shot domain generalization, which combines the challenges of few-shot classification and domain generalization.

**Conditional Batch Normalization** de Vries et al. (2017) proposed conditional batch normalization to modulate visual processing by predicting the scalars $\gamma$ and $\beta$ of the batch normalization conditioned on the language from an early processing stage. Conditional batch normalization has also been applied to align different data distributions for domain adaptation (Li et al., 2016). Oreshkin et al. (2018) applies conditional batch normalization to metric-based models for the few-shot classification task. Tseng et al. (2020) proposed a learning-to-learn method to optimize the hyper-parameters of the feature-wise transformation layers by conditional batch normalization for cross-domain classification. Unlike conditional batch normalization, we use extra data (the query set) to generate normalization statistics under the meta-learning setting, rather than the scalars.

## 3 METHODOLOGY

We view finding appropriate statistics for batch normalization as a density estimation problem. We need to infer the distribution parameters, such as, $\mu$ and $\sigma$ when a Gaussian distribution is presumed, as in existing batch normalization approaches. The motivation behind MetaNorm is to leverage the meta-learning setting and learn from data to generate adaptive normalization statistics. MetaNorm is generic and model-agnostic, addressing batch normalization in a unified way for different settings by minimizing the KL divergence, which is a common metric to measure the difference between two

probability distributions:

$$D_{\mathrm{KL}}\big[q_\phi(m)|p_\theta(m)\big], \tag{3}$$

where $m$ is a random variable that represents the distribution of activations, $p_\theta(m)$ and $q_\phi(m)$ are defined as Gaussian distributions with different implementations depending on the task of interest, e.g., few-shot classification or domain generalization. We leverage the amortized inference technique (Kingma & Welling, 2013) and implement this by inference networks. To be more specific, for each individual channel in each $\ell$ convolutional layer, we infer the moments $\mu$ and $\sigma$ by $f_\mu^\ell(\cdot)$ and $f_\sigma^\ell(\cdot)$, respectively, which are realized as multi-layer perceptrons and we call hypernetworks (Ha et al., 2016). Hypernetworks use one network to generate the weights for another network. Our hypernetworks generate the statistics from data by using amortization techniques.

We simply incorporate the $D_{\mathrm{KL}}$ term into the optimization of the existing model with the cross-entropy loss $\mathcal{L}_{\mathrm{CE}}$, resulting in a general loss function as follows:

$$\mathcal{L} = \mathcal{L}_{\mathrm{CE}} - \lambda D_{\mathrm{KL}}\big[q_\phi(m)|p_\theta(m)\big] \tag{4}$$

where $\lambda > 0$ is a regularization hyper-parameter.

**MetaNorm for Few-Shot Classification**    In the few-shot classification scenario, we define the $\mathcal{C}$-way $K$-shot problem using the episodic formulation from (Vinyals et al., 2016). Each task $\mathcal{T}_i$ is a classification problem sampled from a task distribution $p(\mathcal{T})$. The tasks are divided into a *training meta-set* $\mathcal{T}^{tr}$, *validation meta-set* $\mathcal{T}^{val}$, and *test meta-set* $\mathcal{T}^{test}$, each with a disjoint set of target classes (i.e., a class seen during testing is not seen during training). The validation meta-set is used for model selection, and the testing meta-set is used only for final evaluation. Each task instance $\mathcal{T}_i \sim p(\mathcal{T})$ is composed of a support set $\mathcal{S}$ and a query set $\mathcal{Q}$, and only contains $N$ classes randomly selected from the appropriate meta-set.

We aim to infer statistics from the support set that better match the query set. Therefore, we adopt a straightforward criterion for the inference:

$$D_{\mathrm{KL}}\big[q_\phi(m|S)||p_\theta(m|Q)\big], \tag{5}$$

where we define $q(m|S)=\mathcal{N}(\mu_S, \sigma_S)$ and $p(m|Q)=\mathcal{N}(\mu_Q, \sigma_Q)$, which are the distributions inferred from the support and query sets in a few-shot learning task. By minimizing the KL term in conjunction with the prime objective of a meta-learning algorithm, we are able to find the appropriate statistics from limited data samples for batch normalization. The KL term adheres to a closed form, which makes it easy to implement and computationally efficient. The $p(m|Q)$ can be estimated by directly calculating statistics using the query set, which however performs inferior to inference by optimization. We note the inference from the query set only happens during meta-training time and we use the learned inference network to generate normalization statistics at meta-test time for a test task using its support set.

To infer $\mu_S$, we deploy an inference function $f_\mu^\ell(\cdot)$ that takes activations of a sample as input, and the outputs from all samples are then averaged as the final $\mu_S$:

$$\mu_S = \frac{1}{|\mathcal{S}|} \sum_{i=1}^{|\mathcal{S}|} f_\mu^\ell(\mathbf{a}_i), \tag{6}$$

where $\mathbf{a}_i \in \mathbb{R}^{w \times h}$ is the flattened vector of the activation map of the $i$-th sample in the support set, $w$ is the width of activations, and $h$ is the height of the activation map. To infer $\sigma_S$, we use the obtained $\mu_S$ and deploy a separate inference function $f_\sigma^\ell(\cdot)$:

$$\sigma_S = \frac{1}{|\mathcal{S}|} \sum_{i=1}^{|\mathcal{S}|} f_\sigma^\ell\big((\mathbf{a}_i - \mu_S)^2\big). \tag{7}$$

It is worth mentioning that we actually use each sample to infer the statistics and take the average of all inferred statistics as the final normalization statistics. This enables us to fully exploit the samples to generate more accurate statistics.

Note that the inference functions $f_\mu^\ell(\cdot)$ and $f_\sigma^\ell(\cdot)$ are shared by different channels in the same layer and we will learn $L$ pairs of those functions if we have $L$ convolutional layers in the meta-learning

model. They are parameterized by feed-forward multiple layer perception networks, which we call hypernetworks. Using these hypernetworks, we generate support moments $(\mu_S, \sigma_S)$ and query moments $(\mu_Q, \sigma_Q)$ from the support and query sets, which are used for calculating the KL term in Eq. (5) for optimization during meta-training time. At meta-training time, we apply the statistics inferred from the support set for normalization of both support and query samples:

$$a' = \gamma \left( \frac{a - \mu_S}{\sqrt{\sigma_S^2 + \epsilon}} \right) + \beta, \tag{8}$$

where $\gamma$ and $\beta$ are jointly learned with parameters of the hypernetworks at meta-training time and directly applied at meta-test time, as in conventional batch normalization. At meta-test time, given a test task, we use hypernetworks that take the support set as input to generate normalization statistics directly used for the query set.

**MetaNorm for Domain Generalization**  In the domain generalization scenario, we adopt the meta-learning setting from (Li et al., 2018a; Balaji et al., 2018; Du et al., 2020), and divide a dataset into the source domains used for training and the target domains held out for testing. At meta-training time, data in the source domains is episodically divided into sets of meta-source $\mathcal{D}^s$ and meta-target $\mathcal{D}^t$ domains.

In a similar vein to few-shot classification, we would like to learn to acquire the ability to generate domain-specific statistics from a single example, which can then be applied to unseen domains. We assume we can generate reasonable normalization statistics by using only one sample from the new domain, because, intuitively, a single sample already carries sufficient domain information. We use a single example and all the examples in the same domain to infer the domain-specific statistics and minimize the KL term:

$$D_{\mathrm{KL}}\big[q_\phi(m|\mathbf{a}_i)||p_\theta(m|\mathcal{D}^s \backslash \mathbf{a}_i)\big], \tag{9}$$

where we define $q(m|\mathbf{a}_i) = \mathcal{N}(\mu_a, \sigma_a)$, and $p(m|\mathcal{D}^s \backslash \mathbf{a}_i) = \mathcal{N}(\mu_D, \sigma_D)$, which are implemented in a similar way as Eq. (6) and Eq. (7), and $\mathbf{a}_i$ is an example from its own domain $\mathcal{D}^s$. In both the meta-source and meta-target domains, each example is normalized using the statistics generated by itself, like in Eq. (8), in which we make $\gamma$ and $\beta$ shared across all domains. The minimization of the KL term in Eq. (9) is to encourage the model to generate domain-specific statistics for normalization from only a single example. This enables us to generate domain-specific statistics on target domains that are never seen at meta-training time.

In practice, we take the sum of all samples in all source domains as follows:

$$\sum_i^{|\mathcal{D}^s|} \sum_j^{J} D_{\mathrm{KL}}\big[q_\phi(m|\mathbf{a}_i)||p_\theta(m|\mathcal{D}_j^s \backslash \mathbf{a}_i)\big], \tag{10}$$

where $\mathcal{D}_j^s$ denotes the $j$-th of $J$ meta-source domains. The inference networks are first at meta-training time learned and then directly used as examples from the target domain at meta-test time. Note that on the meta-target domain we do not apply the KL term; instead, we simply rely on each example to generate its statistics for normalization.

**MetaNorm for Few-Shot Domain Generalization**  We introduce an even more challenging setting, i.e., few-shot domain generalization, that combines the challenges of both few-shot classification and domain generalization. Specifically, we aim to learn a model from a set of classification tasks, each of which has only a few samples in a support set for training and test the model on tasks in a query set, which are in a different domain from the support set. Like few-shot classification, the label space is not shared between training and testing. Cross-domain few-shot learning has been explored recently by Tseng et al. (2020) and Guo et al. (2020). However, the setting of our few-shot domain generalization is different and considered to be more challenging, as the support and query set are from *different* domains in the meta-test stage and the target domain is also unseen throughout the training stage. An example for the few-shot domain generalization setting is provided in Figure 1.

We divide a dataset into the source domains $\mathcal{S}$ used for training and the target domains $\mathcal{T}$ held out for testing. During training time, data in the source domains $\mathcal{S}$ is episodically divided into sets of meta-train $D^s$ and meta-test $\mathcal{D}^t$ domains. We sample $\mathcal{C}$-way $k$-shot data as the support set from each meta-source domain $\mathcal{D}^s$, where $k$ is the number of labelled examples for each of the $\mathcal{C}$ classes. We

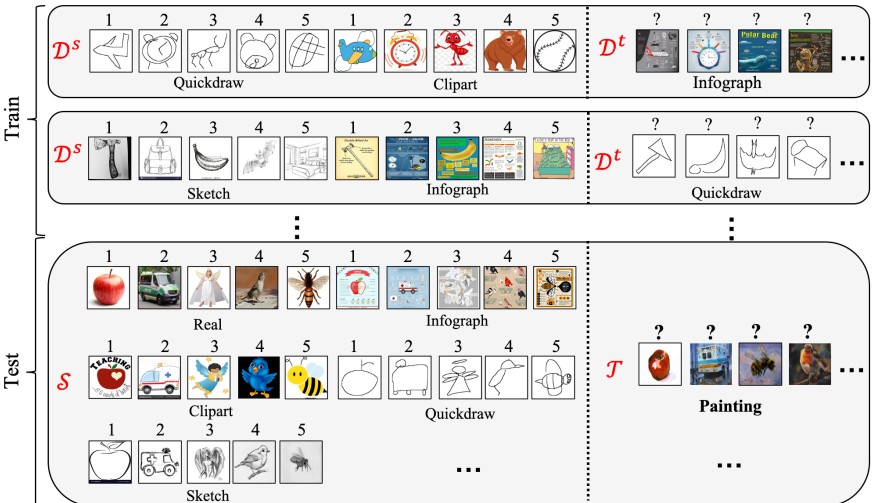

Figure 1: **Illustration of the novel few-shot domain generalization scenario** using the 5-way, 1-shot setting. The training set in the upper box contains the meta-source domains $\mathcal{D}^s$ and the meta-target domain $\mathcal{D}^t$, which are from different domains. Each training task contains meta-source domains with five different classes and one example of each meta-source domain, and more than four examples for evaluation in the meta-target domain. The test set is defined in the same way but with all source domains $\mathcal{S}$ covering classes not present in any of the datasets in the training set, and more than four examples are used for evaluation in the target domain $\mathcal{T}$.

sample $\mathcal{C}$ classes from the meta-test $\mathcal{D}^t$ domain as the query set. At test time, we sample $\mathcal{C}$-way $k$-shot data as the support set from each of the source domains $\mathcal{S}$. The model learned at meta-training time is then fine-tuned on few-shot tasks samples from the source domains and tested on the target domain $\mathcal{T}$. To learn the normalization statistics, we minimize the following KL term:

$$\sum_i^{|\mathcal{D}^s|} D_{\mathrm{KL}}[q_\phi(m|\mathbf{a}_i)||p_\theta(m|\mathcal{D}^s)], \tag{11}$$

where $\mathbf{a}_i$ is the activation associated with each sample from the meta-source domain $\mathcal{D}^s$. Likewise, $q(m|\mathbf{a}_i)$ and $p(m|\mathcal{D}^s)$ are also defined as factorized Gaussian distributions. We also adopt $\gamma$ and $\beta$, which are shared across tasks and jointly learned. MetaNorm learns to acquire the ability to generate proper statistics for itself, and applies it to the samples in the meta-target domain.

## 4 EXPERIMENTAL RESULTS

We conduct an extensive set of experiments on a total of 17 datasets containing more than 15 million images. We use three representative approaches to meta-learning as our base models, i.e., MAML (Finn et al., 2017), ProtoNets (Snell et al., 2017), and VERSA (Gordon et al., 2019), which can verify our MetaNorm is generic, flexible and model-agnostic, making it a simple plug-and-play module that is seamlessly embedded into existing meta-learning approaches. We further compare different normalization methods: transductive batch normalization (TBN), "example" that denotes testing with one example at a time by using TBN, "class" that denotes testing with one class at a time by using TBN, w/o BN which is not using batch normalization, CBN which is using conventional batch normalization, RN (Nichol et al., 2018), MetaBN (Bronskill et al., 2020), TaskNorm-L (Bronskill et al., 2020), and TaskNorm-I (Bronskill et al., 2020). All details about datasets and implementation settings are provided in the appendix. More experimental results, including convergence analysis, are also provided in the appendix. Our code will be publicly released. [1]

**Effect of KL Term** We first conduct ablation studies that measure the effectiveness of MetaNorm. The key of MetaNorm is the introduced KL term for learning to learn statistics. We test the performance of MetaNorm without the KL term by directly using the statistics generated from data.

---

[1]https://github.com/YDU-AI/MetaNorm.

Table 1: **Effect of KL Term** in MetaNorm for few-shot classification with MAML (Finn & Levine, 2018) on *mini*ImageNet and domain generalization on PACS with ResNet-18. More few-shot classification results with ProtoNets (Snell et al., 2017) and VERSA (Gordon et al., 2019), as well as domain generalization results on Office-Home are provided in the appendix. Best performing methods and any other runs within the 95% confidence margin in bold. The KL term is crucial.

| MetaNorm | Few-shot classification | | Domain generalization | | | | |
|---|---|---|---|---|---|---|---|
| | 5-way, 1-shot | 5-way, 5-shot | *Photo* | *Art* | *Cartoon* | *Sketch* | ***Mean*** |
| w/o KL | $34.3 \pm 1.5$ | $50.7 \pm 0.8$ | 88.96 | 71.25 | 65.37 | 69.28 | 73.72 |
| w/ KL | $\mathbf{46.8} \pm \mathbf{1.6}$ | $\mathbf{60.1} \pm \mathbf{0.8}$ | **95.99** | **85.01** | **78.63** | **83.17** | **85.70** |

Figure 2: **Impact of Target Set Size.** The performance increases for larger target sets and plateaus at around 125 for few-shot classification on *mini*ImageNet and around 256 for domain generalization on PACS. TBN here is based on VERSA. MetaNorm generates proper normalization statistics with a reasonable batch size.

In this case, we also use the hypernetworks to generate the moments, $\mu$ and $\sigma$ by simply removing the KL term in the objective function. In Table 1 we present results for few-shot classification on *mini*ImageNet (Vinyals et al., 2016) and for domain generalization on PACS (Li et al., 2017a). The performance of MetaNorm without KL degrades significantly. This is expected, as without the KL term the generation process of normalization statistics lacks direct supervision from the target distribution, resulting in improper statistics.

**Impact of Target Set Size** The other key parameter in MetaNorm is the size of the target set; that is, the number $|\mathcal{Q}|$ of samples in the query set (in few-shot classification) and the number $|\mathcal{D}^s|$ of samples in each domain (in domain generalization). This parameter is important when learning normalization statistics because we use the statistics generated by the target set as the 'ground truth'. We evaluate its impact on the performance of MetaNorm in Figure 2. The experimental results show that TBN is not affected by the target size, both in the 5-way, 1-shot and 5 way, 5-shot tasks. MetaNorm performance rises as the size of the target set increases and plateaus at a reasonable size. In the few-shot setting, the performance reaches its peak at a size of about 125, which is slightly larger than the standard size of 75, while in the domain generalization setting, the performance plateaus at a size of about 128. This demonstrates that we are able to generate proper statistics with the mini-batch gradient descent optimization. In scenarios demanding a very small target set size, we could leverage image synthesis techniques to generate more samples for the targets sets.

**Sensitivity to Algorithm** We evaluate MetaNorm using the MAML (Finn et al., 2017), ProtoNets (Snell et al., 2017) and VERSA (Gordon et al., 2019) algorithms, which are representative gradient, metric and model based meta-learning approaches for few-shot classification. These experiments are conducted on the Omniglot and *mini*ImageNet datasets under different settings. The comparison results on *mini*ImageNet are summarized in Table 2 and the results on Omniglot are provided in the appendix. For all three meta-learning approaches under all settings, MetaNorm consistently achieves comparable performance both to the non-transductive and transductive normalization methods. Being non-transductive, TaskNorm can achieve impressive performance on all the tasks, but its performance is not always better than transductive batch normalization. MetaNorm achieves comparable

Table 2: **Sensitivity to Algorithm.** Few-shot results on *mini*ImageNet using different algorithms. Results on Omniglot are provided in the appendix. Best performing methods and any other runs within the 95% confidence margin in bold. Transductive results indicated above dashed line. MetaNorm is a consistent top-performer, regardless of the meta-learning algorithm.

| | ProtoNets[†] | | MAML[†] | | VERSA[†] | |
|---|---|---|---|---|---|---|
| | 5-way, 1-shot | 5-way, 5-shot | 5-way, 1-shot | 5-way, 5-shot | 5-way, 1-shot | 5-way, 5-shot |
| TBN | $45.9 \pm 0.6$ | $65.5 \pm 0.9$ | $\mathbf{45.5} \pm 1.8$ | $\mathbf{59.7} \pm 0.9$ | $53.4 \pm 1.8$ | $\mathbf{67.3} \pm 0.9$ |
| example | $43.9 \pm 1.9$ | $60.1 \pm 0.8$ | $26.9 \pm 1.5$ | $30.3 \pm 0.7$ | $44.1 \pm 1.7$ | $60.3 \pm 0.7$ |
| class | $43.1 \pm 1.8$ | $59.8 \pm 0.8$ | $26.9 \pm 1.5$ | $27.2 \pm 0.6$ | $43.8 \pm 1.8$ | $59.7 \pm 0.6$ |
| w/o BN | $44.1 \pm 0.5$ | $60.1 \pm 0.6$ | $34.7 \pm 1.5$ | $51.3 \pm 0.8$ | $48.1 \pm 1.5$ | $63.8 \pm 0.6$ |
| CBN (Ioffe & Szegedy, 2015) | $\mathbf{47.8} \pm 0.6$ | $\mathbf{66.7} \pm 0.5$ | $20.1 \pm 0.0$ | $20.2 \pm 0.2$ | $45.7 \pm 1.4$ | $60.7 \pm 0.8$ |
| RN (Nichol et al., 2018) | $39.7 \pm 0.5$ | $63.1 \pm 0.5$ | $40.7 \pm 1.7$ | $57.6 \pm 0.9$ | - | - |
| MetaBN (Bronskill et al., 2020) | $42.6 \pm 0.6$ | $64.6 \pm 0.5$ | $41.6 \pm 1.6$ | $58.6 \pm 0.9$ | $50.1 \pm 1.7$ | $65.8 \pm 0.9$ |
| TaskNorm-L (Bronskill et al., 2020) | $\mathbf{47.5} \pm 0.6$ | $65.3 \pm 0.5$ | $42.0 \pm 1.7$ | $58.1 \pm 0.9$ | $52.1 \pm 1.6$ | $66.1 \pm 0.7$ |
| TaskNorm-I (Bronskill et al., 2020) | $43.2 \pm 0.6$ | $63.9 \pm 0.5$ | $42.4 \pm 1.7$ | $58.7 \pm 0.9$ | $52.9 \pm 1.7$ | $66.5 \pm 0.8$ |
| **MetaNorm** ($|\mathcal{Q}| = 75$) | $47.3 \pm 0.6$ | $65.4 \pm 0.5$ | $44.7 \pm 1.5$ | $\mathbf{59.6} \pm 0.8$ | $52.7 \pm 1.6$ | $\mathbf{67.5} \pm 0.8$ |
| **MetaNorm** ($|\mathcal{Q}| = 125$) | $\mathbf{48.1} \pm 0.6$ | $65.9 \pm 0.9$ | $\mathbf{46.8} \pm 1.6$ | $\mathbf{60.1} \pm 0.8$ | $\mathbf{53.7} \pm 1.6$ | $\mathbf{68.1} \pm 0.8$ |

[†] Results for MAML and ProtoNets (except w/o BN) provided by (Bronskill et al., 2020), and VERSA with TBN provided by (Gordon et al., 2019). All other results based on our re-implementations.

performance to transductive batch normalization, especially under the 5-way-1-shot setting, which is challenging since only a few examples are available to generate statistics. Notice that, MetaNorm performs well with the standard query set size $|\mathcal{Q}|$ of 75 (15 per category). It is slightly better than non-transductive TaskNorm and comparable with TBN. MetaNorm achieves its best performance with a query size $|\mathcal{Q}|$ of 125 (25 per category), only slightly larger than the standard size of 75. This demonstrates the benefit of leveraging meta-learning by MetaNorm for batch normalization. We conclude that MetaNorm is general and serves as a plug-and-play module for existing meta-learning models to improve their performance.

**Sensitivity to Dataset** We evaluate MetaNorm on a demanding few-shot classification challenge called Meta-Dataset (Triantafillou et al., 2020), which is composed of thirteen image classification datasets (eight for training, five testing). To compare with previous work, we perform experiments with ProtoNets and report the results in Table 3. All thirteen per-dataset results can be found in the appendix. MetaNorm achieves high performance in terms of average rank, with highest accuracy on eight of the thirteen datasets. MetaNorm outperforms transductive batch normalization on eleven datasets. It achieves comparable performance with transductive batch normalization on Omniglot and MNIST, which are relatively less challenging. Moreover, MetaNorm performs better than TaskNorm on seven of the thirteen datasets. We conclude that MetaNorm is effective, outperforming alternative normalizations for most datasets.

Table 3: **Sensitivity to Dataset.** Few-shot classification on Meta-Dataset using ProtoNets. MetaNorm performs best overall.

| | Wins | Rank |
|---|---|---|
| IN | – | 10.61 |
| CBN | – | 9.11 |
| LN | 3 | 8.19 |
| TaskNorm-r | – | 7.88 |
| BRN | 1 | 6.23 |
| TBN | 2 | 4.81 |
| MetaBN | 4 | 4.78 |
| RN | 3 | 4.73 |
| TaskNorm-L | 4 | 4.19 |
| TaskNorm-I | 6 | 3.07 |
| **MetaNorm** | **10** | **2.35** |

**Sensitivity to Domains** For this experiment we adopt two widely-used benchmarks for domain generalization of visual object recognition, i.e., PACS (Li et al., 2017a) and Office-Home (Venkateswara et al., 2017). Detailed descriptions on the experimental settings and implementations are provided in the appendix. For fair comparison with prior methods (Balaji et al., 2018; Li et al., 2018b; Seo et al., 2019), we employ ResNet-18 as the backbone network in all experiments. As shown in Table 4, MetaNorm achieves the best performance on PACS and Office-Home in terms of average accuracy. On PACS, MetaNorm consistently outperforms other normalization approaches including domain-specific normalization (Seo et al., 2019), on all four domains. It is worth mentioning that the baseline normalization uses the statistics from the source domains for the batch normalization of the target domain. As expected, the baseline method produces relatively poor performance on most domains, since the source domains cannot provide proper statistics for target domains due to the distribution shift. We have also done an experiment using standard batch normalization. In the training stage, we compute the ground truth statistics using all the test data on the meta-target domain $\mathcal{D}^t$ instead of using inferred statistics $p(m|\mathcal{D}^s \setminus \mathbf{a}_i)$. MetaNorm is still better on most domains and on

Table 4: **Sensitivity to Domains.** Performance comparison on domain generalization. MetaNorm consistently achieves the best performance among all normalization methods.

| | PACS | | | | | Office-Home | | | | |
|---|---|---|---|---|---|---|---|---|---|---|
| | *Photo* | *Art* | *Cartoon* | *Sketch* | *Mean* | *Art* | *Clipart* | *Product* | *Real-World* | *Mean* |
| Baseline | 95.87 | 78.47 | 70.41 | 70.68 | 78.86 | 58.71 | 44.20 | 71.75 | 73.19 | 61.96 |
| IBN (Pan et al., 2018) | 92.04 | 75.29 | 72.95 | 77.42 | 79.43 | 55.41 | 44.82 | 68.28 | 71.95 | 60.09 |
| DSBN (Chang et al., 2019) | 95.51 | 78.61 | 66.17 | 70.15 | 77.61 | 59.04 | 45.02 | 72.67 | 71.98 | 62.18 |
| SN (Luo et al., 2018a) | 93.47 | 82.50 | 76.80 | 80.77 | 83.38 | 54.10 | 44.97 | 64.54 | 71.40 | 58.75 |
| DSON (Seo et al., 2019) | 95.87 | 84.67 | 77.65 | 82.23 | 85.11 | 59.37 | 45.70 | 71.84 | 74.68 | 62.90 |
| Ground truth statistics | 95.78 | **85.17** | 78.15 | 82.91 | 85.50 | 59.35 | **46.12** | 72.77 | 75.08 | 63.33 |
| **MetaNorm** | **95.99** | 85.01 | **78.63** | **83.17** | **85.70** | **59.77** | 45.98 | **73.13** | **75.29** | **63.55** |

average. This is reasonable because ground truth statistics from the test data do not necessarily reflect the true data distribution. The experimental results demonstrate MetaNorm can generate reasonable normalization statistics from only one sample in its domain. We conclude that MetaNorm is effective for domain generalization.

**Few-Shot Domain Generalization**    In our final experiment, we adopt the DomainNet dataset (Peng et al., 2019) and introduce a new, more challenging setting to evaluate the performance for few-shot domain generalization. Detailed descriptions on the dataset and experimental settings are provided in the appendix. We conduct the experiments with the MAML and ProtoNets algorithms under both 5-way 1-shot and 5-way 5-shot settings, and the results are reported in Table 5. We implement transductive batch normalization, MetaBN and the variants of TaskNorm for direct comparison. Under both settings, our MetaNorm produces the best performance and surpasses the transductive batch normalization by large margins of up to 4.0% on the challenging 5-way 1-shot setting with MAML. MetaNorm also achieves better results than the non-transductive TaskNorm approaches. At the same time, with ProtoNet our MetaNorm again consistently delivers the best performance and surpasses both transductive and non-transductive normalizations, The performance on the challenging few-shot domain generalization scenario with different meta-learning algorithms again demonstrates the effectiveness of MetaNorm in handling the challenges of batch normalization for small batches *and* across domains.

Table 5: **Few-Shot Domain Generalization.** Comparison with different normalizations using MAML and ProtoNets on the Few-shot DomainNet dataset. Best performing methods and any other runs within the 95% confidence margin denoted in bold. Reported results use "*Painting*" as the target domain, all based on our implementations. MetaNorm consistently achieves top performance.

| | MAML | | ProtoNets | |
|---|---|---|---|---|
| | 5-way, 1-shot | 5-way, 5-shot | 5-way, 1-shot | 5-way, 5-shot |
| TBN | $28.7 \pm 1.8$ | $49.3 \pm 0.8$ | $27.9 \pm 1.8$ | $47.1 \pm 0.8$ |
| w/o BN | $23.5 \pm 1.7$ | $45.4 \pm 0.7$ | $23.8 \pm 1.8$ | $45.9 \pm 0.7$ |
| CBN | $20.0 \pm 0.0$ | $20.1 \pm 0.2$ | $28.4 \pm 1.8$ | $47.9 \pm 0.7$ |
| MetaBN | $24.7 \pm 1.6$ | $46.1 \pm 0.8$ | $25.1 \pm 1.8$ | $46.1 \pm 0.8$ |
| TaskNorm-L | $26.9 \pm 1.7$ | $47.4 \pm 0.8$ | $\mathbf{29.5} \pm \mathbf{1.6}$ | $\mathbf{48.3} \pm \mathbf{0.8}$ |
| TaskNorm-I | $27.5 \pm 1.6$ | $48.8 \pm 0.6$ | $26.8 \pm 1.8$ | $46.9 \pm 0.7$ |
| **MetaNorm** | $\mathbf{32.7} \pm \mathbf{1.7}$ | $\mathbf{51.9} \pm \mathbf{0.9}$ | $\mathbf{30.7} \pm \mathbf{1.8}$ | $\mathbf{49.1} \pm \mathbf{0.9}$ |

## 5 CONCLUSION

In this paper we present MetaNorm, a meta-learning based batch normalization. MetaNorm tackles the challenging scenarios where the batch size is too small to produce sufficient statistics or when training statistics are not directly applicable to test data due to a domain shift. MetaNorm learns to learn adaptive statistics that are specific to tasks or domains. It is generic and model-agnostic, which enables it to be used with various meta-learning algorithms for different applications. We evaluate MetaNorm on two well-known existing tasks, i.e., few-shot classification and domain generalization, and we also introduce the challenging evaluation scenario of few-shot domain generalization that addresses the small batch and distribution shift problems simultaneously. An extensive evaluation on 17 datasets reveals that MetaNorm consistently achieves results that are better, or at least competitive, compared to other normalization approaches, verifying its effectiveness as a new meta-learning based batch normalization approach.

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

## A ALGORITHMS DESCRIPTIONS

In this Appendix we provide the detailed MetaNorm algorithm descriptions to conduct batch normalization for few-shot classification (Algorithm 1), domain generalization (Algorithm 2) and few-shot domain generalization (Algorithm 3). The dataflow of the implementation is shown in Figure 3.

---

**Algorithm 1** MetaNorm for Few-Shot Classification

---

**Meta-train**: Input values of $a$ over support set $\mathbf{a}_{S,i}$ and query set $\mathbf{a}_{Q,i}$;
$\gamma, \beta \leftarrow$ Initialize parameters.

$\mu_S = \frac{1}{|\mathcal{S}|} \sum_{i=1}^{|\mathcal{S}|} f_\mu^\ell(\mathbf{a}_{S,i}); \mu_Q = \frac{1}{|\mathcal{Q}|} \sum_{i=1}^{|\mathcal{Q}|} f_\mu^\ell(\mathbf{a}_{Q,i});$

$\sigma_S = \frac{1}{|\mathcal{S}|} \sum_{i=1}^{|\mathcal{S}|} f_\sigma^\ell\big((\mathbf{a}_{S,i} - \mu_S)^2\big); \sigma_Q = \frac{1}{|\mathcal{Q}|} \sum_{i=1}^{|\mathcal{Q}|} f_\sigma^\ell\big((\mathbf{a}_{Q,i} - \mu_Q)^2\big);$

$a'_{S,i} = \gamma\left(\frac{a_{S,i}-\mu_S}{\sqrt{\sigma_S^2+\epsilon}}\right) + \beta; a'_{Q,i} = \gamma\left(\frac{a_{Q,i}-\mu_S}{\sqrt{\sigma_S^2+\epsilon}}\right) + \beta;$

$\mathcal{L}_{\text{KL}} = D_{\text{KL}}\big[\mathcal{N}(\mu_\mathcal{S}, \sigma_\mathcal{S}) || \mathcal{N}(\mu_\mathcal{Q}, \sigma_\mathcal{Q})\big]$

**return** $a'_{S,i} = \textbf{MetaNorm}(\mathbf{a}_{S,i}); a'_{Q,i} = \textbf{MetaNorm}(\mathbf{a}_{Q,i}); \mathcal{L}_{KL}$

---

**Meta-test**: Input values of $a$ over support set $\mathbf{a}_{S,i}$ and query set $\mathbf{a}_{Q,i}$;

$\mu_S = \frac{1}{|\mathcal{S}|} \sum_{i=1}^{|\mathcal{S}|} f_\mu^\ell(\mathbf{a}_{S,i});$

$\sigma_S = \frac{1}{|\mathcal{S}|} \sum_{i=1}^{|\mathcal{S}|} f_\sigma^\ell\big((\mathbf{a}_{S,i} - \mu_S)^2\big);$

$a'_{Q,i} = \gamma\left(\frac{\mathbf{a}_{Q,i}-\mu_S}{\sqrt{\sigma_S^2+\epsilon}}\right) + \beta;$

**return** $a'_{Q,i} = \textbf{MetaNorm}(a_{Q,i})$

---

---

**Algorithm 2** MetaNorm for Domain Generalization

---

**Train**: Input values of $a$ over meta-source domain $\mathbf{a}_{S,i}$ and meta-target domain $\mathbf{a}_{T,i}$;
$\gamma, \beta \leftarrow$ Initialize parameters.

$\mu_{S,i} = f_\mu^\ell(\mathbf{a}_{S,i}); \mu_S = \frac{1}{|\mathcal{S}|} \sum_{i=1}^{|\mathcal{S}|} f_\mu^\ell(\mathbf{a}_{S,i});$

$\sigma_{S,i} = f_\sigma^\ell\big((\mathbf{a}_{S,i} - \mu_{S,i})^2\big); \sigma_S = \frac{1}{|\mathcal{S}|} \sum_{i=1}^{|\mathcal{S}|} f_\sigma^\ell\big((\mathbf{a}_{S,i} - \mu_S)^2\big);$

$\mu_{T,i} = f_\mu^\ell(\mathbf{a}_{T,i}); \sigma_{T,i} = f_\sigma^\ell\big((\mathbf{a}_{T,i} - \mu_{T,i})^2\big);$

$a'_{T,i} = \gamma\left(\frac{\mathbf{a}_{T,i}-\mu_{T,i}}{\sqrt{\sigma_{T,i}^2+\epsilon}}\right) + \beta;$

$\mathcal{L}_{KL} = D_{\text{KL}}\big[\mathcal{N}(\mu_{\mathcal{S},\rangle}, \sigma_{\mathcal{S},\rangle}) || \mathcal{N}(\mu_\mathcal{S}, \sigma_\mathcal{S})\big]$

**return** $a'_{T,i} = \textbf{MetaNorm}(a_{T,i}); \mathcal{L}_{\text{KL}}$

---

**Test**: Input values of $a$ over test domain $\mathbf{a}_i$;

$\mu_i = f_\mu^\ell(\mathbf{a}_i);$

$\sigma_i = f_\sigma^\ell\big((\mathbf{a}_i - \mu_i)^2\big);$

$a'_i = \gamma\left(\frac{\mathbf{a}_i-\mu_i}{\sqrt{\sigma_i^2+\epsilon}}\right) + \beta;$

**return** $a'_i = \textbf{MetaNorm}(a_i)$

---

## B DATASETS

We conduct an extensive set of experiments on a total of 17 datasets containing more than 15 million images. All dataset details and settings are provided in this Appendix.

*mini*ImageNet. The *mini*ImageNet is originally proposed in (Vinyals et al., 2016) and has been widely used for evaluating few-shot learning algorithms. It consists of 60,000 color images from 100

---

**Algorithm 3** MetaNorm for Few-Shot Domain Generalization

---

**Meta-train**: Input values of $a$ over meta-source domain set $\mathbf{a}_{S,i}$ and meta-target domain set $\mathbf{a}_{Q,i}$; $\gamma, \beta \leftarrow$ Initialize parameters.

$\mu_S = \frac{1}{|\mathcal{S}|} \sum\limits_{i=1}^{|\mathcal{S}|} f_\mu^\ell(\mathbf{a}_{S,i}); \mu_Q = \frac{1}{|\mathcal{Q}|} \sum\limits_{i=1}^{|\mathcal{Q}|} f_\mu^\ell(\mathbf{a}_{Q,i});$

$\sigma_S = \frac{1}{|\mathcal{S}|} \sum\limits_{i=1}^{|\mathcal{S}|} f_\sigma^\ell\big((\mathbf{a}_{S,i} - \mu_S)^2\big); \sigma_Q = \frac{1}{|\mathcal{Q}|} \sum\limits_{i=1}^{|\mathcal{Q}|} f_\sigma^\ell\big((\mathbf{a}_{Q,i} - \mu_Q)^2\big);$

$a'_{S,i} = \gamma\left(\frac{a_{S,i} - \mu_S}{\sqrt{\sigma_S^2 + \epsilon}}\right) + \beta; a'_{Q,i} = \gamma\left(\frac{a_{Q,i} - \mu_S}{\sqrt{\sigma_S^2 + \epsilon}}\right) + \beta;$

$\mathcal{L}_{\mathrm{KL}} = D_{\mathrm{KL}}\big[\mathcal{N}(\mu_\mathcal{S}, \sigma_\mathcal{S}) || \mathcal{N}(\mu_\mathcal{Q}, \sigma_\mathcal{Q})\big]$

**return** $a'_{S,i} = \textbf{MetaNorm}(\mathbf{a}_{S,i}); a'_{Q,i} = \textbf{MetaNorm}(\mathbf{a}_{Q,i}); \mathcal{L}_{KL}$

---

**Meta-test**: Input values of $a$ over support set $\mathbf{a}_{S,i}$ and query set $\mathbf{a}_{Q,i}$;

$\mu_S = \frac{1}{|\mathcal{S}|} \sum\limits_{i=1}^{|\mathcal{S}|} f_\mu^\ell(\mathbf{a}_{S,i});$

$\sigma_S = \frac{1}{|\mathcal{S}|} \sum\limits_{i=1}^{|\mathcal{S}|} f_\sigma^\ell\big((\mathbf{a}_{S,i} - \mu_S)^2\big);$

$a'_{Q,i} = \gamma\left(\frac{\mathbf{a}_{Q,i} - \mu_S}{\sqrt{\sigma_S^2 + \epsilon}}\right) + \beta;$

**return** $a'_{Q,i} = \textbf{MetaNorm}(a_{Q,i})$

---

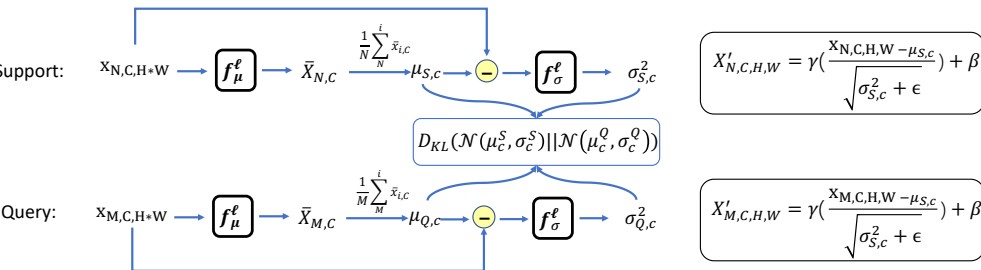

Figure 3: **The dataflow of the implementation for few-shot learning.** "N" indicates support size, "M" indicates query size, "C" indicates the channel of activations, "W" indicates the width of activations, "H" indicates the height of activations.

classes with 600 examples per class. The images have dimensions of $84 \times 84$ pixels. We follow the train/val/ test split introduced in (Ravi & Larochelle, 2017), which uses 64 classes for meta-training, 16 classes for meta-validation, and the remaining 20 classes for meta-testing.

**Omniglot**. Omniglot (Lake et al., 2015) is a few-shot learning dataset consisting of 1,623 handwritten characters (each with 20 instances) derived from 50 alphabets. We follow the pre-processing and training procedure defined in (Vinyals et al., 2016). We resize images to $28 \times 28$. The training, validation and test sets consist of a random split of 1,100, 100, and 423 characters.

**PACS** (Li et al., 2017a) contains a total of 9,991 images of the size $224 \times 224$ from 4 domains, i.e., *photo*, *art-painting*, *cartoon* and *sketch*, which demonstrate huge domain gaps. Images are from 7 object classes, i.e., *dog*, *elephant*, *giraffe*, *guitar*, *horse*, *house*, and *person*. We follow the "leave-one-out" protocol in (Li et al., 2017a; 2018b; Carlucci et al., 2019), where the model is trained on any three of the four domains, which we call source domains, and tested on the last (target) domain. The train-val-test splits are the same as in (Li et al., 2017a).

**Office-Home** (Venkateswara et al., 2017) also has 4 domains: *art*, *product*, *clipart* and *real-world*. For each domain, the dataset contains images of 65 object categories found typically in office and home settings. We use the same experimental protocol as for PACS.

**DomainNet** (Peng et al., 2019) contains 6 distinct domains, i.e., *clipart*, *infograph*, *painting*, *quick-draw*, *real*, and *sketch* for 345 categories. The categories are from 24 divisions, which are: *Furniture*, *Mammal*, *Tool*, *Cloth*, *Electricity*, *Building*, *Office*, *Human Baby*, *Road Transportation*, *Food*, *Nature*,

*Cold Blooded*, *Music*, *Fruit*, *Sport*, *Tree*, *Bird*, *Vegetable*, *Shape*, *Kitchen*, *Water Transportation*, *Sky Transportation*, *Insect*, *Others*.

**Meta-Dataset** (Triantafillou et al., 2020) is composed of ten (eight train, two test) existing image classification datasets. These are: *ILSVRC-2012* (ImageNet, (Russakovsky et al., 2015)), *Omniglot* (Lake et al., 2015), *Aircraft* (Maji et al., 2013), *CUB-200-2011* (Birds, (Wah et al., 2011)), *Describable Textures* (Cimpoi et al., 2014), *Quick Draw*, *Fungi*, *VGG Flowr* (Nilsback & Zisserman, 2008), *Traffic Signs* (Houben et al., 2013) and *MSCOCO* (Lin et al., 2014). Each episode generated in Meta-Dataset uses classes from a single dataset. Two of these datasets, *Traffic Signs* and *MSCOCO*, are fully reserved for evaluation, it means no classes from these sets participate in the training set. Except for *Traffic Signs* and *MSCOCO*, the remaining datasets contribute some classes to each of training, validation and test splits of classes. There are about 14 million images in total in Meta-Dataset.

## C  FEW-SHOT DOMAINNET

To construct Few-shot DomainNet, we chose 200 random classes from DomainNet and used 140 for training, 20 for validation and the last 40 for testing. Note that the last 40 object classes were never seen during training. The dataset consists of 200,000 colour images of size $84 \times 84$ with each of the 200 classes having 1,000 examples. Please see Table 6, Table 7 and Table 8 for training, validation, and test classes.

Table 6: **Training classes of Few-shot DomainNet**

| |
|---|
| *Furniture*: bathtub, ceiling fan, couch, fence, hot tub, mailbox |
| *Mammal*: tiger, rhinoceros, bat, cat, lion, panda |
| *Tool*: anvil, basket, broom, sword, pliers |
| *Cloth*: belt, camouflage, eyeglasses, crown, bowtie |
| *Electricity*: calculator, computer, camera, cooler, dishwasher |
| *Building*: bridge, jail, pool, tent, castle |
| *Office*: alarm clock, binoculars, backpack, book, bandage |
| *Human Body*: arm, ear, face, beard, elbow, finger, brain, eye, foot, knee |
| *Road Transportation*: ambulance, bus motorbike, bicycle, train |
| *Food*: birthday cake, cookie, hot dog, peanut, sandwich, bread, donut, pizza, steak, lollipop |
| *Nature*: beach, lightning, ocean, river, sun, cloud, moon, rain, tornado |
| *Cold Blooded*: crab, frog, crocodile, lobster, fish, octopus, shark |
| *Music*: cello, guitar, saxophone, violin, clarinet, harp, trombone |
| *Fruit*: apple, banana, blackberry, blueberry, grapes, pear |
| *Sport*: baseball, baseball bat, basketball, snorkel, yoga, tennis racquet |
| *Tree*: bush, grass, cactus, tree, flower |
| *Bird*: bird, owl |
| *Vegetable*: asparagus, broccoli, carrot, mushroom, onion |
| *Shape*: circle, hexagon |
| *Kitchen*: fork, frying pan, hourglass, knife, lighter |
| *Water Transportation*: aircraft carrier, canoe, cruise ship, submarine |
| *Sky Transportation*: airplane, helicopter |
| *Insect*: ant, bee |
| *Others*: angel, cannon, dragon, mermaid, stop sign, snowman, feather |

Table 7: **Validation classes of Few-shot DomainNet**

| |
|---|
| *Furniture*: stairs, ladder |
| *Mammal*: monkey |
| *Tool*: paint can |
| *Cloth*: purse, t-shirt |
| *Electricity*: radio |
| *Building*: pond |
| *Office*: nail |
| *Human Body*: skull, tooth |
| *Road Transportation*: firetruck |
| *Food*: - |
| *Nature*: star, hurricane |
| *Cold Blooded*: sea turtle |
| *Music*: - |
| *Fruit*: strawberry |
| *Sport*: hockey stick |
| *Tree*: - |
| *Bird*: penguin |
| *Vegetable*: - |
| *Shape*: - |
| *Kitchen*: wine bottle |
| *Water Transportation*: - |
| *Sky Transportation*: - |
| *Insect*: -- |
| *Others*: teddy-bear |

## D    IMPLEMENTATION DETAILS.

In the few-shot learning task, MAML and ProtoNets use a simple CNN containing 4 convolutional layers, each of which is a $3 \times 3$ convolution with 32 filters, followed by MetaNorm, a ReLU non-linearity, and finally a $2 \times 2$ max-pooling. VERSA uses a CNN containing 5 convolutional layers, each of which is a $3 \times 3$ convolution with 64 filters, followed by MetaNorm, a ReLU non-linearity, and finally a $2 \times 2$ max-pooling. In the domain generalization task, we rely on ResNet-18 as backbone for fair comparison with previous work. Each convolutional layer is followed by MetaNorm. The hypernetwork is a 3-layer MLP with 128 units per layer and rectifier nonlinearities. We implemented all models in the Tensorflow framework and tested on an NVIDIA Tesla V100. All code will be available at: https://github.com/YDU-AI/MetaNorm.

### D.1    MAML EXPERIMENTS

For MAML experiments, we used the codebase by Finn (Finn, 2017). We use the Adam optimizer with default parameters, and a meta batch size of 4 tasks. The number of test episodes is set as 600. The number of training iterations is 60,000. We set $\lambda=0.001$. The other hyper-parameters we use are the default MAML parameters. No early stopping was used. We used the first-order approximation of MAML for the experiments.

Table 8: **Test classes of Few-shot DomainNet**

| |
|---|
| *Furniture*: teapot, toothpaste, stove, umbrella |
| *Mammal*: mouse, |
| *Tool*: bucket, paint can |
| *Cloth*: sweater, shoe, flip flops |
| *Electricity*: television, stereo, toaster, flashlight |
| *Building*: waterslide, garden |
| *Office*: map, clock, calendar, scissors |
| *Human Body*: finger, nose, toe |
| *Road Transportation*: bulldozer |
| *Food*: peanut |
| *Nature*: mountain, sun |
| *Cold Blooded*: lobster, scorpion |
| *Music*: harp |
| *Fruit*: pineapple |
| *Sport*: soccer ball, hockey stick |
| *Tree*: house plant, leaf |
| *Bird*: swan |
| *Vegetable*: string bean |
| *Shape*: squiggle |
| *Kitchen*: - |
| *Water Transportation*: - |
| *Sky Transportation*: - |
| *Insect*: – |
| *Others*: feather, snowman |

## D.2 PROTONETS EXPERIMENTS

For ProtoNets, we used the codebase by Fatir (Fatir, 2018). For *mini*ImageNet, we used the following ProtoNets options: a learning rate of 0.001, 60,000 training iterations, 200 validation episodes, 600 test episodes and $\lambda$=0.0001. We choose the units of hidden layers and $\lambda$ by cross-validation. For Meta-Dataset, we reproduce the code provided by CNAPS (Requeima et al., 2019) with TensorFlow. We simply replace its normalization method with our MetaNorm method and add the KL term to the final loss. We are consistent with the dataset configuration and follow the training process as specified in (Triantafillou et al., 2020). The number of training iterations is 80,000. We use a constant learning rate of 0.0001. We set $\lambda$=0.001. We follow TaskNorm's (Bronskill et al., 2020) options: they do not use feature adaptation, and allow updates pre-trained feature extractor weights during meta-training stage.

## D.3 VERSA EXPERIMENTS

For VERSA, we used the codebase by Gordon (Gordon, 2019). For the 5-way 5-shot model, we train using the setting of 8 tasks per batch for 100,000 iterations and use a constant learning rate of 0.0001, $\lambda$= 0.001. For the 5-way 1-shot model, we train with the setting of 8 tasks per batch for 150,000 iterations and use a constant learning rate of 0.00025, $\lambda$=0.01. We set validation episodes as 200, and test episodes as 600. The units of hidden layers and $\lambda$ were chosen by cross-validation.

Table 9: Inference function $f_\mu^l(\cdot)$

| Output size | Layers |
|---|---|
| $w \times h$ | Input flattened vector of the activation map |
| 128 | fully connected, ELU |
| 128 | fully connected, ELU |
| $w \times h$ | fully connected to $\mu$ |

Table 10: Inference function $f_\sigma^l(\cdot)$

| Output size | Layers |
|---|---|
| $w \times h$ | Input flattened vector of the activation map and $\mu$ |
| 128 | fully connected, ELU |
| 128 | fully connected, ELU |
| $w \times h$ | fully connected to $\sigma$ |

# E    EXTRA RESULTS FOR EFFECT OF KL TERM

In this Appendix we consider extra results for the ablation on measuring the effect of the KL term. We report results for few-shot classification on *mini*ImageNet with ProtoNets (Snell et al., 2017) and VERSA (Gordon et al., 2019) in Table 11. We also report domain generalization results on Office-Home in Table 12. In all cases the KL term is crucial.

Table 11: **Effect of KL Term** in MetaNorm for few-shot classification on *mini*ImageNet with ProtoNets and VERSA. Best performing methods and any other runs within 95% confidence margin denoted in bold.

| | ProtoNets | | VERSA | |
|---|---|---|---|---|
| **MetaNorm** | 5-way, 1-shot | 5-way, 5-shot | 5-way, 1-shot | 5-way, 5-shot |
| w/o KL | $40.1 \pm 1.6$ | $58.7 \pm 0.8$ | $48.7 \pm 1.6$ | $64.3 \pm 0.8$ |
| w/ KL | $\mathbf{48.1} \pm \mathbf{1.6}$ | $\mathbf{65.9} \pm \mathbf{0.9}$ | $\mathbf{53.7} \pm \mathbf{1.6}$ | $\mathbf{68.1} \pm \mathbf{0.8}$ |

Table 12: **Effect of KL Term** in MetaNorm for domain generalization on Office-Home.

| | Office-Home | | | | |
|---|---|---|---|---|---|
| **MetaNorm** | *Art* | *Clipart* | *Product* | *Real-World* | *Mean* |
| w/o KL | 51.25 | 39.27 | 69.75 | 68.19 | 57.12 |
| w/ KL | **59.77** | **45.98** | **73.13** | **75.29** | **63.55** |

# F    SENSITIVITY TO ALGORITHM ON OMNIGLOT

The experiments on Omniglot for few-shot classification under the meta-learning settings of MAML, VERSA and ProtoNets are reported in Tables 13, 14 and 15. MetaNorm consistently outperforms both transductive and non-transductive normalization approaches.

# G    SENSITIVITY TO DATASET

The complete set of results for each of the thirteen datasets in Meta-Dataset are provided in Table 16.

Table 13: **Sensitivity to Algorithm.** Few-shot results on Omniglot using MAML. Best performing methods and any other runs within the 95% confidence margin in bold. Transductive results indicated above dashed line.

| | Omniglot[†] | | | |
|---|---|---|---|---|
| | 5-way, 1-shot | 5-way, 5-shot | 20-way, 1-shot | 20-way, 5-shot |
| TBN | **98.4** $\pm$ **0.7** | **99.2** $\pm$ **0.2** | **90.9** $\pm$ **0.5** | 96.6 $\pm$ 0.2 |
| example | 21.6 $\pm$ 1.3 | 22.0 $\pm$ 0.5 | 3.7 $\pm$ 0.2 | 5.5 $\pm$ 0.2 |
| class | 21.6 $\pm$ 1.3 | 23.2 $\pm$ 0.5 | 3.7 $\pm$ 0.2 | 14.5 $\pm$ 0.3 |
| w/o BN | 92.6 $\pm$ 0.9 | 90.7 $\pm$ 0.1 | 84.3 $\pm$ 0.4 | 91.7 $\pm$ 0.2 |
| CBN (Ioffe & Szegedy, 2015) | 20.1 $\pm$ 0.0 | 20.0 $\pm$ 0.0 | 5.0 $\pm$ 0.0 | 5.0 $\pm$ 0.0 |
| RN (Nichol et al., 2018) | 92.6 $\pm$ 0.9 | 98.2 $\pm$ 0.2 | 89.0 $\pm$ 0.6 | 96.8 $\pm$ 0.2 |
| MetaBN (Bronskill et al., 2020) | 91.8 $\pm$ 0.9 | 98.1 $\pm$ 0.3 | 89.6 $\pm$ 0.5 | 96.4 $\pm$ 0.2 |
| TaskNorm-L (Bronskill et al., 2020) | 94.0 $\pm$ 0.8 | 98.0 $\pm$ 0.3 | 89.6 $\pm$ 0.5 | 96.4 $\pm$ 0.2 |
| TaskNorm-I (Bronskill et al., 2020) | 94.4 $\pm$ 0.8 | 98.6 $\pm$ 0.2 | 90.0 $\pm$ 0.5 | 96.3 $\pm$ 0.2 |
| **MetaNorm** | **98.8** $\pm$ **0.5** | **99.3** $\pm$ **0.2** | **91.3** $\pm$ **0.5** | **97.1** $\pm$ **0.2** |

[†] Results (except w/o BN and our MetaNorm) provided by (Bronskill et al., 2020).

Table 14: **Sensitivity to Algorithm.** Few-shot results on Omniglot using VERSA. Best performing methods and any other runs within the 95% confidence margin in bold. Transductive results indicated above dashed line.

| | Omniglot[†] | | | |
|---|---|---|---|---|
| | 5-way, 1-shot | 5-way, 5-shot | 20-way, 1-shot | 20-way, 5-shot |
| TBN | **99.7** $\pm$ **0.2** | **99.8** $\pm$ **0.2** | **97.7** $\pm$ **0.2** | **98.8** $\pm$ **0.1** |
| example | 94.9 $\pm$ 0.2 | 95.1 $\pm$ 0.3 | 92.9 $\pm$ 0.2 | 95.9 $\pm$ 0.2 |
| class | 94.3 $\pm$ 0.3 | 94.8 $\pm$ 0.1 | 91.8 $\pm$ 0.3 | 95.1 $\pm$ 0.4 |
| w/o BN | 95.6 $\pm$ 0.7 | 96.5 $\pm$ 0.1 | 93.1 $\pm$ 0.3 | 96.3 $\pm$ 0.2 |
| CBN (Ioffe & Szegedy, 2015) | 94.3 $\pm$ 0.3 | 95.7 $\pm$ 0.0 | 92.7 $\pm$ 0.2 | 95.2 $\pm$ 0.3 |
| MetaBN (Bronskill et al., 2020) | 96.7 $\pm$ 0.3 | 98.1 $\pm$ 0.3 | 95.8 $\pm$ 0.2 | 97.1 $\pm$ 0.2 |
| TaskNorm-L (Bronskill et al., 2020) | 97.9 $\pm$ 0.3 | 99.2 $\pm$ 0.2 | 96.1 $\pm$ 0.2 | 98.0 $\pm$ 0.2 |
| TaskNorm-I (Bronskill et al., 2020) | 98.3 $\pm$ 0.2 | 99.5 $\pm$ 0.2 | 96.7 $\pm$ 0.2 | 98.1 $\pm$ 0.1 |
| **MetaNorm** | **99.8** $\pm$ **0.1** | **99.9** $\pm$ **0.1** | **97.9** $\pm$ **0.2** | **98.8** $\pm$ **0.2** |

[†] Results of TBN provided by (Gordon et al., 2019). All other results based on our re-implementations.

Table 15: **Sensitivity to Algorithm.** Few-shot results on Omniglot using ProtoNets. Best performing methods and any other runs within 95% confidence margin denoted in bold. Transductive results indicated above dashed line.

| | Omniglot[†] | | | |
|---|---|---|---|---|
| | 5-way, 1-shot | 5-way, 5-shot | 20-way, 1-shot | 20-way, 5-shot |
| TBN | 98.4 $\pm$ 0.2 | **99.6** $\pm$ **0.2** | 94.5 $\pm$ 0.2 | **98.6** $\pm$ **0.1** |
| example | 98.4 $\pm$ 0.2 | **99.5** $\pm$ **0.2** | 94.3 $\pm$ 0.2 | 98.5 $\pm$ 0.1 |
| class | 98.4 $\pm$ 0.2 | **99.3** $\pm$ **0.2** | 94.2 $\pm$ 0.2 | 98.4 $\pm$ 0.1 |
| w/o BN | 94.6 $\pm$ 0.7 | 95.5 $\pm$ 0.1 | 91.7 $\pm$ 0.3 | 94.3 $\pm$ 0.2 |
| CBN (Ioffe & Szegedy, 2015) | 98.5 $\pm$ 0.2 | **99.6** $\pm$ **0.1** | 94.5 $\pm$ 0.2 | **98.6** $\pm$ **0.1** |
| RN (Bronskill et al., 2020) | 98.0 $\pm$ 0.2 | **99.6** $\pm$ **0.1** | 94.1 $\pm$ 0.2 | **98.6** $\pm$ **0.1** |
| MetaBN (Bronskill et al., 2020) | 98.4 $\pm$ 0.2 | **99.6** $\pm$ **0.1** | 94.5 $\pm$ 0.2 | **98.6** $\pm$ **0.1** |
| TaskNorm-L (Bronskill et al., 2020) | 98.6 $\pm$ 0.2 | **99.6** $\pm$ **0.1** | 95.0 $\pm$ 0.2 | **98.7** $\pm$ **0.1** |
| TaskNorm-I (Bronskill et al., 2020) | 98.4 $\pm$ 0.2 | **99.6** $\pm$ **0.2** | 93.4 $\pm$ 0.2 | **98.6** $\pm$ **0.1** |
| **MetaNorm** | **98.9** $\pm$ **0.2** | **99.7** $\pm$ **0.2** | **95.8** $\pm$ **0.2** | **98.9** $\pm$ **0.2** |

[†] Results (except w/o BN and our MetaNorm) provided by (Bronskill et al., 2020).

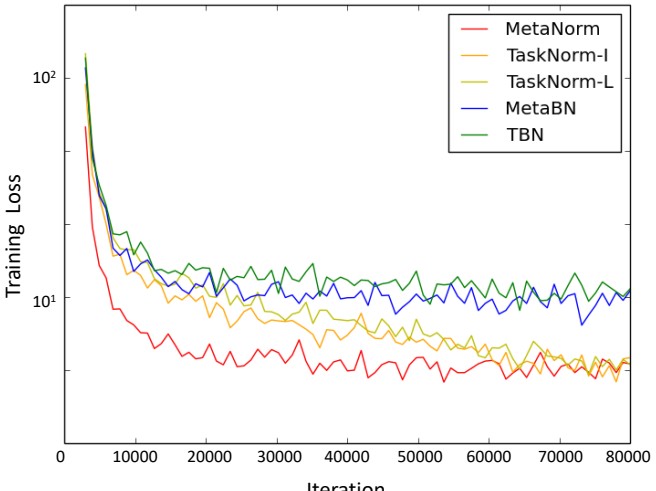

Figure 4: **Training loss.** Results of using the ProtoNets algorithm on *mini*ImageNet with respect to training loss versus iterations. Our MetaNorm achieves fastest training convergence.

## H    TRAINING SPEED

We plot the training loss versus training iterations by using the ProtoNets algorithm in Figure 4. MetaNorm achieves fastest training convergence. From Table 2 and Figure 4, MetaNorm achieves best classification accuracy and training efficiency, which demonstrates the benefit of leveraging meta-learning by MetaNorm for batch normalization.

Table 16: **Sensitivity to Dataset.** Few-shot classification results on Meta-Dataset using ProtoNets. The ± sign indicates the 95% confidence interval over tasks. Best performing methods and any other runs within the 95% confidence margin in bold. Results of other methods provided by Bronskill et al. (2020).

| | ILSVRC | Omniglot | Aircraft | Birds | Textures | Quick Draw | Fungi | VGG Flower | Traffic Signs | MSCOCO | MNIST | CIFAR10 | CIFAR100 | Rank |
|---|---|---|---|---|---|---|---|---|---|---|---|---|---|---|
| TBN | **44.7 ± 1.0** | **90.7 ± 0.6** | 83.3 ± 0.6 | 69.6 ± 0.9 | 61.2 ± 0.7 | 75.0 ± 0.8 | 46.4 ± 1.0 | 83.1 ± 0.6 | 64.0 ± 0.8 | 38.2 ± 1.0 | 93.4 ± 0.4 | 64.7 ± 0.8 | 48.0 ± 1.1 | 4.81 |
| CBN (Ioffe & Szegedy, 2015) | 43.6 ± 1.0 | 77.5 ± 1.1 | 77.0 ± 0.7 | 67.5 ± 0.9 | 57.7 ± 0.7 | 62.1 ± 1.0 | 43.6 ± 1.0 | 82.3 ± 0.6 | 59.5 ± 0.8 | 36.6 ± 1.0 | 86.5 ± 0.6 | 57.3 ± 0.8 | 43.1 ± 1.0 | 9.11 |
| BRN (Ioffe, 2017) | 43.0 ± 1.0 | 89.1 ± 0.7 | **84.4 ± 0.5** | 69.0 ± 0.9 | 58.0 ± 0.7 | 74.3 ± 0.8 | 46.5 ± 1.0 | 84.5 ± 0.6 | 65.7 ± 0.8 | 38.4 ± 1.0 | 91.9 ± 0.4 | 60.1 ± 0.8 | 43.9 ± 1.0 | 6.23 |
| LN (Ba et al., 2016) | 33.9 ± 0.9 | **90.8 ± 0.6** | 73.9 ± 0.7 | 54.1 ± 1.0 | 55.8 ± 0.8 | 72.5 ± 0.8 | 33.2 ± 1.1 | 78.3 ± 0.8 | **69.1 ± 0.7** | 30.1 ± 0.9 | **94.0 ± 0.4** | 51.5 ± 0.8 | 34.0 ± 0.9 | 8.19 |
| IN (Ulyanov et al., 2016) | 32.5 ± 0.9 | 83.4 ± 0.8 | 75.0 ± 0.6 | 50.2 ± 1.0 | 45.3 ± 0.7 | 70.8 ± 0.8 | 29.8 ± 1.0 | 69.4 ± 0.8 | 60.7 ± 0.8 | 27.7 ± 0.9 | 87.4 ± 0.5 | 50.5 ± 0.8 | 32.1 ± 1.0 | 10.61 |
| RN (Nichol et al., 2018) | **45.1 ± 1.0** | **90.8 ± 0.6** | 80.9 ± 0.6 | 68.6 ± 0.9 | 64.1 ± 0.7 | 75.4 ± 0.7 | 46.7 ± 1.0 | 84.4 ± 0.7 | 66.0 ± 0.8 | 37.3 ± 1.0 | **93.9 ± 0.4** | 62.3 ± 0.8 | 47.2 ± 1.1 | 4.73 |
| MetaBN (Bronskill et al., 2020) | 44.2 ± 1.0 | **90.4 ± 0.6** | 82.3 ± 0.6 | 68.6 ± 0.8 | 60.5 ± 0.7 | 74.2 ± 0.7 | 46.5 ± 1.0 | **86.0 ± 0.6** | 63.2 ± 0.8 | **38.6 ± 1.1** | **93.9 ± 0.4** | 63.0 ± 0.8 | 47.0 ± 1.0 | 4.78 |
| TaskNorm-r (Bronskill et al., 2020) | 42.7 ± 1.0 | 88.6 ± 0.7 | 79.6 ± 0.6 | 64.2 ± 0.9 | 60.8 ± 0.7 | 73.2 ± 0.8 | 42.3 ± 1.1 | 81.1 ± 0.7 | 64.9 ± 0.8 | 35.4 ± 1.0 | 92.5 ± 0.4 | 61.4 ± 0.8 | 45.2 ± 1.0 | 7.58 |
| TaskNorm-L (Bronskill et al., 2020) | **45.1 ± 1.1** | **90.2 ± 0.6** | 81.2 ± 0.6 | 68.8 ± 0.9 | 63.4 ± 0.8 | 75.4 ± 0.7 | 46.5 ± 1.0 | 82.9 ± 0.7 | 67.0 ± 0.7 | **39.2 ± 1.0** | 91.9 ± 0.4 | **66.9 ± 0.8** | 51.3 ± 1.1 | 4.09 |
| TaskNorm-I (Bronskill et al., 2020) | **44.9 ± 1.0** | **90.6 ± 0.6** | **84.7 ± 0.5** | **71.0 ± 0.9** | 65.9 ± 0.7 | **77.5 ± 0.7** | 49.6 ± 1.1 | 83.2 ± 0.6 | 65.8 ± 0.7 | 38.5 ± 1.0 | 93.3 ± 0.4 | **67.6 ± 0.8** | 50.0 ± 1.0 | 3.07 |
| MetaNorm | 45.3 ± 1.0 | **90.8 ± 0.5** | 83.3 ± 0.6 | **70.6 ± 0.8** | **66.7 ± 0.6** | **77.6 ± 0.6** | **51.1 ± 0.6** | **86.3 ± 0.7** | 68.1 ± 0.6 | **39.7 ± 0.9** | 92.1 ± 0.5 | **67.1 ± 0.7** | **52.7 ± 0.9** | **2.35** |

Table 17: **Effect of number of units of hidden layers** in MetaNorm for few-shot classification with MAML (Finn & Levine, 2018) on *mini*ImageNet. The $\pm$ sign indicates the 95% confidence interval over tasks. We achieve best results with 128 units of hidden layers.

| | MAML | |
|---|---|---|
| | 5-way, 1-shot | 5-way, 5-shot |
| $n = 64$ | $44.3 \pm 1.5$ | $58.1 \pm 0.8$ |
| $n = 128$ | $46.8 \pm 1.6$ | $60.1 \pm 0.8$ |
| $n = 256$ | $46.2 \pm 1.6$ | $59.8 \pm 0.8$ |
| $n = 512$ | $45.9 \pm 1.5$ | $59.5 \pm 0.9$ |
| $n = 1024$ | $44.9 \pm 1.5$ | $58.7 \pm 0.8$ |

