# OpenReview forum: "MetaNorm: Learning to Normalize Few-Shot Batches Across Domains"
_ICLR.cc/2021/Conference — ICLR 2021 Poster_

### Official Review · AnonReviewer2 · 2020-10-20
**Review of "MetaNorm: Learning to Normalize Few-Shot Batches Across Domains"**

**Rating:** 4
**Confidence:** 5

**Review:**

This paper describes a new method for normalizing few-shot learning episodes. The authors point out that the statistics of an episode are unreliable when the size of the episode is small or when the data distribution changes from episode to episode. To remedy this, the authors propose a method called ‘MetaNorm’ which uses a meta-learning approach to infer the means and variances to be used in the batch normalization layers that are employed in the feature extractor component. In particular, they meta-learn the parameters for a set of hypernetworks in an amortized fashion that learn to generate the means and variances of the batch normalization layers conditioned on the contents of the episode. The paper focuses entirely on the few-shot image classification scenario where MetaNorm is evaluated in various settings including standard few-shot classification and domain generalization (including a novel few-shot domain generalization setting).

**Pros:**
Fundamentally, the concept behind MetaNorm is innovative and promising. The idea of using meta-learned hypernetworks to generate the means and variances of the normalization layers is good. The paper also makes a good case for the effectiveness of a KL term that is added to the classifer loss function that helps the hypernetworks learn a good set of parameters. The range of experiments is sufficient.

**Cons:**
The paper is poorly executed and missing too much information to ascertain that the method outperforms competitive approaches.

Specific concerns:
(1) There is no mention of the MetaNorm training efficiency (which is generally understood to be the primary goal of batch normalization, see [1,2]). How does training efficiency of MetaNorm compare to other methods? And if training efficiency is not a goal, MetaNorm should be compared to methods whose goal is to adapt a classification system to a variety of data distributions (e.g. [3,4,5,6, etc.]).

(2) Reproducibility: The paper is currently missing details that are required to reproduce the results and ensure fair comparison with other methods including:
- Which existing codebases were used (if any)?
- What hyper-parameters and training settings are used? (e.g. learning rates, number of training iterations, the use of early stopping, the number of test episodes, weight decay, etc.)
- In Section 4, “Impact of Target Set Size” indicates that the number of samples in the query set is a key parameter for MetaNorm and that 125 for the few-shot scenario and 128 for the domain generalization scenario are the best values to use. Are these the values used in the experiments? If not, what values are used?

(3) There are many omissions in the paper:
- The variable ‘m’ as used in equations 3, 4, 8, 9, 10 is not defined.
- The KL expressions in equations 4, 8, and 10 should be written as a proper minimization expression where the parameters that are minimized are explicitly stated.
- The total loss function for MetaNorm is never stated. One could assume that it is a sum between the regular loss function for the particular few shot learning problem and the KL term, but it should be explicitly stated. Also, is there a weight on the KL term in the loss function? If so, how is the optimal weight determined?
- Various methods in the tables are not defined or referenced including RN, TaskNorm-I, TaskNorm-L, ‘class’, and ‘example’.
- How are the confidence intervals on the results in the table calculated? Are they 95% confidence intervals, or something else?
- In the tables, how do you decide which entry to ‘bold’ given the confidence intervals? In table 2, TBN (and sometimes TaskNorm) are often within error bars of MetaNorm but are not bolded. Same for tables 11 and 13, 14.
- It would be good to have more information on the hypernetworks that generate the normalization parameters. In Section 2, it says: “They are three layer networks with one hidden layer of 128 units and the input size depends on the feature dimensions of the convolutional layers.” How big is the other hidden layer and output size of the network? What sort of non-linearities are used, etc.?
- In section 2 it says: “Note that the inference functions $f_\mu^l()$ and $f_\sigma^l()$ are shared by different channels in the same layer and we will learn L pairs of those functions if we have L convolutional layers in the meta-learning model”. What is the justification for this parameterization (say versus having unique functions per channel)? Was an ablation study that compares various configurations done?

(4) There are factual errors in the paper:
- In Section 1 and Section 3, the authors state that Prototypical networks uses transductive batch normalization. This is not the case (refer to https://github.com/jakesnell/prototypical-networks for the version of Prototypical networks authored by Jake Snell). Please correct.
- In section 1 it states that TaskNorm “remains under-performing compared to transductive batch normalization”. While the results listed in Table 2 support this statement, the results in Table 3 refute this statement. Please clarify.

(5) There are several missing attributions and some exact text taken from other papers that is not quoted or cited:
- In Table 2, the competitive results for MAML and ProtoNets seem to be extracted from [2], yet these are not attributed. Did you reproduce them? Were the MetaNorm numbers generated with a similar code base and hyper-parameters?
- In Tables 11, 13, and 14, the competitive results also seem to be extracted from [2] and were not attributed. Were they reproduced, or just stated? Were the MetaNorm numbers generated with a similar code base and hyper-parameters?
- The following sentences seem to be directly copied from [2]. If so, there should be quotation marks and attribution:
“Batch normalization relies on the implicit assumption that the dataset comprises i.i.d. samples from some underlying distribution.”
“The challenge constructs few-shot learning tasks by drawing from the following distribution. First, one of the datasets is sampled uniformly; second, the “way” and “shot” are sampled randomly according to a fixed procedure; third, the classes and context / target instances are sampled.”
“In the meta-test phase, the identity of the original dataset is not revealed and the tasks must be treated independently (i.e. no information can be transferred between them). The meta-training set comprises a disjoint and dissimilar set of classes from those used for meta-test.”

(6) In Figure 2, the title says “Impact of target set size.”. However, for the left and middle plot, the horizontal axis is labeled as |S|, which is the support set size. Please resolve the ambiguity.
Minor Comments:
- In section 2, MetaNorm for Few-Shot Classification: The sentence “The $p(m|Q)$ can be estimated by directly calculating statistics using the query set, which however performs inferior to inference by optimization.” is grammatically incorrect.
- In section 2, MetaNorm for Few-Shot Classification: should “multiple layer perception networks” be “multi-layer perceptron networks”? Similarly, in section 2 it says: “…which are realized as multi-layer perceptions and we call hypernetworks.” ‘perceptions’ should be ‘perceptrons’.
- Sometimes the nomenclature support set / query set is used, and other places it refers to the same as context set / target set. Please use consistent nomenclature.
- In section 2, it says “…which are realized as multi-layer perceptions and we call hypernetworks”. And later in section 2, it says: “They are parameterized by feed-forward multiple layer perception networks, which we call hypernetworks”.  Note that hypernetworks are an established concept, you should cite [7].
- Section 1 should clarify in the batch normalization exposition that the methods described in the paper apply only to normalizing 2D convolutional layers (i.e. does not apply to fully connected layers as they do not have feature maps).

**References:**
[1] Sergey Ioffe and Christian Szegedy. Batch normalization: Accelerating deep network training by reducing internal covariate shift. In International Conference on Machine Learning, pp. 448–456, 2015.
[2] John Bronskill, Jonathan Gordon, James Requeima, Sebastian Nowozin, and Richard Turner. Tasknorm: Rethinking batch normalization for meta-learning. In International Conference on Machine Learning. 2020.
[3] Rebuffi, Sylvestre-Alvise, Hakan Bilen, and Andrea Vedaldi. "Learning multiple visual domains with residual adapters." Advances in Neural Information Processing Systems. 2017.
[4] Rebuffi, Sylvestre-Alvise, Hakan Bilen, and Andrea Vedaldi. "Efficient parametrization of multi-domain deep neural networks." Proceedings of the IEEE Conference on Computer Vision and Pattern Recognition. 2018.
[5] Requeima, James, et al. "Fast and flexible multi-task classification using conditional neural adaptive processes." Advances in Neural Information Processing Systems. 2019.
[6] Tseng, Hung-Yu, et al. "Cross-domain few-shot classification via learned feature-wise transformation." arXiv preprint arXiv:2001.08735 (2020).
[7] Ha, David, Andrew Dai, and Quoc V. Le. "Hypernetworks." arXiv preprint arXiv:1609.09106 (2016).

---

> ### Author Response · Authors · 2020-11-23
> **Response to AnonReviewer2**
>
> We thank *AnonReviewer2* for the thorough review and detailed constructive comments.
>
> **(1) There is no mention of the MetaNorm training efficiency (which is generally understood to be the primary goal of batch normalization, see [1,2]). How does training efficiency of MetaNorm compare to other methods?**
>
> The reviewer is right. We plot the training loss versus training iterations with the ProtoNets algorithm (the codebase we use here is based on [Fatir's](https://github.com/abdulfatir/prototypical-networks-tensorflow) re-implementation.) as shown in Figure 3 in the appendix. MetaNorm achieves the fastest training convergence, demonstrating the benefit of leveraging meta-learning by MetaNorm for batch normalization. We have added the "Training Loss versus iteration" in the appendix. Thank you.
>
> **(2) Reproducibility: The paper is currently missing details that are required to reproduce the results and ensure fair comparison with other methods including:**
>
> * We clarify our codebases in the appendix. For MAML we use the code provided by [Finn](https://github.com/cbfinn/maml). For ProtoNets, we use the code provided by [Fatir](https://github.com/abdulfatir/prototypical-networks-tensorflow). For VERSA,  we use the code provided by [Gordon](https://github.com/Gordonjo/versa). For TaskNorm,  we reproduce it ourselves based on the above three models and [TaskNorm](https://github.com/cambridge-mlg/cnaps). All models are implemented with TensorFlow. Our code will be publicly released.
>
> * Different models have different training settings. The details of the experimental setting have been added in the appendix as well.
>
> * For a fair comparison, we always choose 125 query samples for the few-shot scenario and 128 target samples for the domain generalization.
>
> **(3) There are many omissions in the paper:**
>
> Thank you for your detailed guidance. Much appreciated.
>
> * We have added after equation 3 that "m" is a random variable that represents the distribution of activation.
>
> * We have rewritten the KL expressions.
>
> * We have added the total loss in the revised version of the paper. The weight of the KL term of the different models is different. We have explained them in the appendix of the experimental setting. The references and descriptions of the various methods have been added in the revised version.
>
> * Indeed, the $\pm$ indicates the 95\% conﬁdence interval over tasks in the few-shot learning. We have added this description in the updated paper.
>
> * We make bold according to the highest value without considering the following error bar. We have added this description in the revised version to avoid confusion. It is worth noting that, we take the error bar into account when computing the average ranking.
>
> * We choose the units of the hidden layers by cross-validation. We implemented the hypernetworks with $64$, $128$, $256$, $512$ units. We found $128$ can get the best performance. We have added this experiment in the appendix. We use elu as non-linearities, and the output layer does not use any non-linearities. We also have added the structure of inference networks in the appendix.
>
> * Yes, we did an ablation study on various configurations. We found that shared inference functions are better than those not shared across channels. This would be because the way of sharing parameters by different channels can reduce network parameters and increase computational efficiency. We have added this in the appendix.
>
> **(4) There are factual errors in the paper:**
>
>  * We have corrected it. Thank you.
>
> *  We have modified this sentence as "its performance is not always performing better than transductive batch normalization".
>
> **(5) There are several missing attributions and some exact text taken from other papers that is not quoted or cited:**
>
> We apologize for hasty and imprecise writing. We fixed it in the revised version.
>
> * We have clarified what numbers in Tables 2, 11, 13, and 14 are provided by [Bronskill et al. (2020)](https://arxiv.org/pdf/2003.03284.pdf) and which ones have been computed by us.
>
> * MetaNorm is based on the different codebases of each model and uses different hyper-parameters.
>
> * The indicated sentences have been re-phrased in the updated version. Thank you.
>
> **(6)  In Figure 2, the title says "Impact of target set size.". However, for the left and middle plot, the horizontal axis is labeled as |S|, which is the support set size. Please resolve the ambiguity. Minor Comments:**
>
> All repaired. Thank you.

---

> > ### Comment · AnonReviewer2 · 2020-11-23
> > **Thanks for the updated version of the paper! A few more questions...**
> >
> > - In many of the charts, the errors bar are not taken into consideration when choosing which methods to bold. What is the rationale for not taking the error bars into account? Looking at Table 2, if error bars are taken into account, the claim that MetaNorm consistently outperforms transductive and non-transductive approaches is false. You can say it is comparable to TBN, but does not outperform. Same comment for Tables 13, 14, and 15.
> >
> > - In your response to the review, you indicate that "We always choose 125 query samples for the few-shot scenario and 128 target samples for the domain generalization." However, in Tables 2, 13, 15, you use results directly from Bronskill et al. and in that implementation they use far smaller target sets, putting those implementations at a disadvantage and making the comparison unfair. Increasing the target set size normally results in better classification accuracy and training convergence as more data is used in the task. Ideally, you would use the same parameters as the competitive methods for comparison purposes and then in a separate part of the chart, state your current numbers using the larger query set. How do the results change if you use comparable parameters?
> >
> > - No implementation details (codebase, hyperparameters, etc.) have been provided for the Meta-Dataset experiments. Can you provide them? Was the same large target size also used for the MetaNorm Meta-Datset experiments?
> > - Why did you evaluate on only 200 test tasks? The norm for most benchmarks (Omniglot, miniImageNet, Meta-Dataset) is 600 test tasks. Were your results not as good when evaluated on more tasks?
> >
> > - In table 2, for Versa, you quote the TBN accuracies from Gordon et al., but this has not been acknowledged in the table. Also, for the 5-way, 1-shot Versa experiments, you used 3 times as many training iterations as the Gordon et al implementation. You should provide a comparison using similar hyper-parameters.
> >
> > - The only training plot provided was for ProtoNets. How did the training convergence compare for Omniglot and Meta-Dataset? Was the same Target Size used for all methods when plotting Figure 4?
> >
> > - In Table 16, you should check your ranking code as it does not seem to be compatible with the algorithm used by Triantafillou et al.
> >
> > Typos:
> >
> > - 'activation' should be'activations' in the sentence '$m$ is a random variable that represents the distribution of activation'.
> >
> > - grammatical error in the sentence 'Hypernetworks are using one network', it should be 'Hypernetworks use one network'
> >
> > - missing the word 'is' before 'a' in the clause 'where $\lambda > 0$ a regularization hyper-parameter.'
> >
> > - Protonets is 'Snell et al', not 'Snake et al'
> >
> > - Impact of Target Set Size section: It mentions the results are in Figure 4, should that be Figure 2?

---

> > > ### Author Response · Authors · 2020-11-24
> > > **Second response to AnonReviewer2**
> > >
> > > We are indebted to *AnonReviewer2*  for pushing the quality and precision of our manuscript forward.
> > >
> > > * The reviewer is right. We have taken the error bar into account in Table 2, 13, 14 and 15, re-bolded the data in the tables, and explained accordingly. We also softened our conclusion in Table 2: ``MetaNorm is a consistent top-performer, regardless of the meta-learning algorithm.'' And in the corresponding paragraph: "MetaNorm achieves comparable performance to transductive batch normalization, especially under the 5-way-1-shot setting, which is challenging since only a few examples are available to generate statistics."
> > >
> > > * Our MetaNorm only increases the query size during meta-training, allowing our model to learn the ability to infer the statistics of the test task using only a few samples. In the meta-test stage, we use the query size consistent with the previous methods, which makes the comparison fair to our opinion.  We observe from Figure 2 that TBN does not seem to be affected by the target size, under both the 5-way, 1-shot, and 5 way, 5-shot settings. Due to the time limit, we are unable to complete all experiments with non-transductive methods at the same large query size during the rebuttal phase, we certainly would like to add these experiments in the final version. We ask for your understanding and trust.
> > >
> > > * We have added the implementation details of Meta-Dataset in appendix D.2. We use the same large target size for our MetaNorm Meta-Dataset experiments.
> > >
> > > * Indeed, we evaluate 200 validation tasks, 600 test tasks following exactly the same setting of [VERSA](https://github.com/Gordonjo/versa). We apologize for this mistake and we have fixed it.
> > >
> > > *  The results of VERSA with TBN are reproduced by us using the author-provided code.
> > > Compared with VERSA, we have the extra KL-term in our final loss, which needs more iterations to converge. For the comparison, we also have increased the iterations for VERSA. We found the performance of VERSA does not increase after convergence, even with more iterations. We believe a comparison based on the convergence, instead of the same number of iterations, to be fair as well.
> > >
> > > * We will add training convergence comparisons on Omniglot and Meta-Dataset in the final version. We clarify that the other methods in Figure 4 use the query size from their original implementations ([TaskNorm](https://github.com/cambridge-mlg/cnaps) and [ProtoNets](https://github.com/abdulfatir/prototypical-networks-tensorflow)).
> > >
> > > * An error bar was dropped while copying, we have fixed it.  Thank you.
> > >
> > > All identified typos have been fixed. We did another pass over the manuscript leading to a few more repairs. Thanks again.

---

> > > > ### Comment · AnonReviewer2 · 2020-11-24
> > > > **A few more remarks**
> > > >
> > > > **Small things:**
> > > >
> > > > - In Table 3, the number of wins for each method do not correspond to the results in Table 16. Please fix.
> > > > - In the first paragraph in Section D, you say: "We implemented all models in the Tensorflow framework...", but in section D.2, you indicate that you used the code by the CNAPS authors and when I looked at that code, it was implemented primarily in PyTorch. Please fix.
> > > >
> > > > **The larger issue:**
> > > >
> > > > - My primary reservation about MetaNorm is that it requires an unreasonably large target set at training time. In realistic few-shot learning scenarios, that quantity of labeled training data is simply not available, preventing the use of MetaNorm in those situations. This limitation is not given enough attention in the paper i.e. there was no attempt to acknowledge it as a limitation and address the issue in any way (e.g. generate additional target set data with an image synthesis technique such as a GAN, etc.).
> > > > - Secondly, I am concerned that the comparisons made in the tables and figures are not 'apples to apples'. Ideally all experiments would be carried out at the standard target set size and the large target set size and then conclusions drawn. The performance improvements thus far demonstrated by MetaNorm are not enormous and it is unclear if those performance gains would hold up if the methods were evaluated under similar conditions. At very least, there should be some indication in the tables and figures that the target size used during training differs among competitive methods.

---

> > > > > ### Author Response · Authors · 2020-11-25
> > > > > **Third response to AnonReviewer2**
> > > > >
> > > > > **Small things:**
> > > > >
> > > > > * Thank you. When taking the error bar into account in Table 3, other methods become more competitive and MetaNorm maintains the best average rank with the highest accuracy on eight of the thirteen datasets. We have adapted the text describing the experiment accordingly.
> > > > >
> > > > > *  We clarify that we reproduce the PyTorch code provided by CNAPS with TensorFlow for Meta-Dataset. All our (re-)implemented TensorFlow code will be released.
> > > > >
> > > > > **The larger issue:**
> > > > >
> > > > > * We thank *AnonReviewer2* for sharing the primary reservation. We respectfully disagree on MetaNorm requiring an unreasonable large target set. Table 18 shows MetaNorm performs well with the standard query set size of 75 (15 per category). It is slightly better than TaskNorm and comparable with TBN. MetaNorm achieves its best performance with a query size of 125 (25 per category), only slightly larger than the standard size of 75. We do not consider this increase unreasonable, nor impractical. We do agree it deserves more attention and we have added a discussion in the subsection "Impact of Target Set Size", including your compelling suggestion to leverage image synthesis techniques.
> > > > >
> > > > > *  We believe our Meta-Dataset comparison in Table 16 is 'apples to apples', as different methods are evaluated on similar conditions. We have also conducted the experiments on miniImageNet with the standard query set size using MAML and VERSA. The results are shown in Table 18 of the appendix. MetaNorm achieves comparable performance to non-transductive approaches, e.g., TaskNorm, under similar conditions, and the performance is very close to that of TBN. We conclude the performance gains are predominantly from the proposed meta-learning batch normalization mechanism, rather than simply a larger query set. We would like to mention that TBN *always* needs to use the entire query set, even during the meta-test stage, while MetaNorm does not rely on the query set anymore at meta-test. A fact that we believe to be a considerable advantage.
> > > > >
> > > > >  We thank the reviewer.

---

> > > > > > ### Comment · AnonReviewer2 · 2020-11-25
> > > > > > **Thanks for the additional information**
> > > > > >
> > > > > > Thank you for adding Table 18. All my concerns about making fair comparisons would be alleviated if the results in all of the relevant tables and figures were reported for both the standard and expanded target set sizes. For example, in the final revision of the paper, it would be great if the additional information in Table 18 was added as a new section to Table 2 and similarly for the other tables and figures where there is disparity in the target set sizes between the methods.
> > > > > >
> > > > > > Table 18 indeed shows that MetaNorm performance is within error bars of TBN and TaskNorm at the standard target set size of 75, which is in itself a worthy result. The fact that MetaNorm performance increases with larger target set sizes is a bonus.

---

> > > > > > > ### Author Response · Authors · 2020-11-25
> > > > > > > **Fourth response to AnonReviewer2**
> > > > > > >
> > > > > > > Thank you. We will include results for standard and expanded target set sizes, as suggested, in the final version.

---

### Official Review · AnonReviewer1 · 2020-10-28
**Official Blind Review #1**

**Rating:** 7
**Confidence:** 3

**Review:**

This paper proposes to replace batch normalization statistics, which are typically computed as the batch moments during training or a fixed training average during testing, with the outputs of learned neural networks. These networks are trained to minimize the KL divergence between their output and the expected or desired batch statistics. In this way, the statistics computation is amortized and can hopefully generalize in the face of small batches and distribution shift.

Pros:

+ The paper is generally well written and easy to follow.
+ There are extensive experiments demonstrating the empirical effectiveness of the proposed approach.

Cons:

- Perhaps missing some additional details that would help the reader better understand why and how the method works.

Currently, I recommend acceptance because I believe the pros rather handily outweigh the cons. I provide more details below.


Quality
---

The paper is well written and the experiments are thorough and well executed. Related to the con listed above, a few additional experiments could be useful for gaining a deeper understanding of the method.

First, why do we expect an amortized procedure to produce reasonable normalization statistics when tested "out of distribution", e.g., on a new domain? It would be interesting to report, during testing, are the inferred statistics actually close to the ground truth statistics on the test domain? Or otherwise, are they interpretable in some other fashion? Any other insight that the authors could supply regarding this general question would also be appreciated.

Second, a similar question can be asked about the fully "in distribution" setting, i.e., where batch normalization was invented. If one were to apply the proposed method in a standard supervised learning setting (e.g., minimize KL between inferred statistics on each training point and the ground truth statistics computed on the training batch), would the inferred test time statistics come close to either the average statistics computed through training or perhaps even the ground truth test statistics? Is it any worse than standard batch norm? Could it perhaps even be better?

Clarity
---

Most of my concerns about clarity, which are small, relate to the experiments suggested above, i.e., better exposition of how and why the method works. Providing intuition throughout the paper regarding this point would strengthen the paper.

The few shot domain generalization setting is not too difficult to understand if the reader is familiar with both the few shot learning and domain generalization settings individually -- it is probably difficult to parse otherwise, but that is perhaps hard to rectify given the space constraints.

There also seem to be a few instances of "meta" vs "non meta" terms being misused, e.g., "training" vs "meta-training" when describing domain generalization.

Originality
---

As far as I am aware, this work presents a novel method, though I am not an expert regarding the relevant prior work. The citations to batch normalization as used in domain adaptation seem appropriate. A few other papers in a similar vein perhaps should also be cited:

https://arxiv.org/abs/1603.04779
https://arxiv.org/abs/2002.04019
https://arxiv.org/abs/2006.10963
https://arxiv.org/abs/2006.16971

So although this general line of work seems to be attracting a decent amount of attention, this work is still novel in that it amortizes the inference process and presents extensive empirical results.

Significance
---

This work seems significant to researchers interested in meta-learning, domain generalization, and problems involving distribution shift in general. The proposed method is also relatively simple, allowing for easier adoption and further testing.

---

> ### Author Response · Authors · 2020-11-23
> **Response to AnonReviewer1**
>
> We thank *AnonReviewer1*  for the insightful comments and the acknowledgment of the originality and significance of this work.
>
> **Perhaps missing some additional details that would help the reader better understand why and how the method works. [...]
> First, why do we expect an amortized procedure to produce reasonable normalization statistics when tested out of distribution, e.g., on a new domain? It would be interesting to report, during testing.
> Second, a similar question can be asked about the fully "in distribution" setting, i.e., where batch normalization was invented. If one were to apply the proposed method in a standard supervised learning setting (e.g., minimize KL between inferred statistics on each training point and the ground truth statistics computed on the training batch), would the inferred test time statistics come close to either the average statistics computed through training or perhaps even the ground truth test statistics? Is it any worse than standard batch norm? Could it perhaps even be better?**
>
> Thank you for sharing the insight. We assume we can generate reasonable normalization statistics by using only one sample from the new domain, because, intuitively, a single sample already carries much of its domain information. This ability is indeed acquired by meta-learning. We have done the suggested experiment using standard batch normalization. In the training stage, we compute the ground truth statistics using all the test data on the meta-test domain $\mathcal{D}^t$ instead of using the inferred statistics $p(m|\mathcal{D}^{s}\backslash \mathbf{a}_i)$. Results in Table 4 reveal MetaNorm is still better on most domains and on average. This is reasonable because ground truth statistics from the test data do not necessarily reflect the true data distribution. The experimental results demonstrate MetaNorm can generate reasonable normalization statistics from only one sample in its domain. We have added the discussion to the *Sensitivity to Domains* subsection.
>
> **Most of my concerns about clarity, which are small, relate to the experiments suggested above, i.e., better exposition of how and why the method works. Providing intuition throughout the paper regarding this point would strengthen the paper.**
>
> We have added the intuition about "how and why the method works", and provided a new comparison for domain generalization by using ground truth statistics in the revised version of the paper.
>
> **There also seem to be a few instances of "meta" vs "non meta" terms being misused, e.g., "training" vs "metatraining" when describing domain generalization.**
>
> The description of domain generalization has been fixed.
>
> **As far as I am aware, this work presents a novel method, though I am not an expert regarding the relevant prior work. The citations to batch normalization as used in domain adaptation seem appropriate. A few other papers in a similar vein perhaps should also be cited.**
>
> We have expanded the related work section, and discuss the suggested references in *related work*. Thank you.

---

### Official Review · AnonReviewer3 · 2020-10-28
**This work address batch normalization by introduce a KL term for learning to learn statistic**

**Rating:** 6
**Confidence:** 3

**Review:**

Summary: The paper proposes an effective meta-learning normalization, named MetaNorm, to infer adaptive statistics for batch normalization by minimizing the KL divergence. The module is lightweight. The proposed method is evaluated on few-shot learning, domain generalization, and few-shot domain generalization.

Justification of rating:  Overall, the proposed approach is logical and sound. The formulation and methodology seem to be correct. As I am not personally working on the specific topics, I might not be able to discover major issues in this work.

Strengths:
+ The proposed MetaNorm leverage meta-learning approach with KL divergence for learning to learn normalization statistics from data.
+ The proposed approach is model agnostic and can be easily embedded into meta-learning approaches.
+ Evaluation demonstrates it is able to learn with few examples, as well as handle variations in domain for domain generalization problem. The experimental results show consistent improvement over compared baselines (Table 2-5)
+ The paper provide sufficient information for researcher to reproduce the results. Code will be released in the future.

Minor comments:
- The implementation of the hypernetworks is effectively MLP with one hidden layers of 128 units. Has the author conduct ablation on hypernetwork on various configuration?

---

> ### Author Response · Authors · 2020-11-23
> **Response to AnonReviewer3**
>
> We thank *AnonReviewer3* for the honest assessment.
>
> **The implementation of the hypernetworks is effectively MLP with one hidden layers of 128 units. Has the author conduct ablation on hypernetwork on various configuration?**
>
> Indeed, we implemented the hypernetworks with $64$, $128$, $256$, $512$ units. We choose the units of the hidden layers by cross-validation. We found $128$ obtains the best performance. We have added this ablation in the appendix. Thank you.

---

### Official Review · AnonReviewer4 · 2020-10-28
**Interesting approach, some improvements to the paper (experiments) needed**

**Rating:** 6
**Confidence:** 4

**Review:**

The authors propose a method for cross domain few-shot classification that learns to generate domain specific data statistics from very few training examples for domain independent batch normalisation.
They propose to train small auxiliary networks that generate data statistics for normalisation. Networks are trained within a meta-learning framework using a KL divergence loss, which enforces estimated statistics on small support/training examples to match statistics from query sets where more data is available.


STRENGTHS

The paper is well written and motivated. The proposed approach is, for the most part, easy to follow and understand. The approach benefits from its simplicity and versatility (e.g. it is not tied to a specific FSL method) and promising performance is obtained.

The authors provide a very large set of experiments in multiple scenarios, and definitely demonstrates a strong effort in evaluating their approach.


WEAKNESSES

Unfortunately, despite the fact that a significant amount of time was dedicated to evaluate the method, the experimental section needs substantial modifications. This is mainly due to the presentation of the experiments, as well as some key missing comparisons.
Regarding presentation, too many implementation details and descriptions of the experiments are missing from the main paper (and in certain cases, missing altogether). Regarding datasets and implementation, it is ok to provide non essential details in the appendix (especially considering the large number of datasets considered). However, no information is provided at all in the main paper, which makes it very difficult to understand the setting of the experiments. For example, in Table 1 and Figure 2, it is not known on which datasets experiments are run, and it is never mentioned in the main text (for table 1) what FSL method is employed.
In addition, Table 2 experiments comparing MetaNorm to different approaches sorely lacks description. The approach is compared to 9 different algorithms, none of which are given a description or reference to learn more about the method. It is therefore impossible, besides guessing, to know what the model is being compared to. This issue is also noticeable in Table 3-5, in particular with a baseline in Table 4 that is never described.
Finally, experiments in Figure 2 would be a lot more interesting if compared to standard methods. It would be interesting to see how the proposed strategy allows to be more sample efficient and reach stronger performance than transductive batch norm in situations where sample size is the smallest. As this is one of the cited main limitations of TBN, this experiment is highly important2- With regards to related work, I would suggest to move the section after the introduction, where it provides much better context to facilitate method comprehension, in particular regarding the description of the TBN strategy.

Authors should also comment on how their work, and in particular FSL domain generalisation setting, relates to cross domain few-shot learning works,
e.g. Tseng et al ICLR 2020, Cross-Domain Few-Shot Classification via Learned Feature-Wise Transformation
It is currently presented as a completely new approach to FSL, and appears to ignore past cross domain FSL works. Please provide additional context regarding such works.


RECOMMENDATION

In summary, the authors propose an interesting, simple strategy for more robust BN that can be of interest to the community. I would strongly recommend that the authors make the following modifications to strengthen their paper:

1-	Reorganise and expand on the experimental evaluation to provide necessary details and make the main paper self-contained, even if it requires moving an experiment to the supplementary materials.
2-	Move the related works section after the introduction, to provide additional context before delving into the method
3-	Relate the proposed work to past work on cross-domain FSL and potentially tone down claims that few-shot cross domain learning is a completely new problem investigated here.
4-	Please provide a comparison to standard TBN in Figure 2


Additional suggestions

5-	If possible, please provide an overview figure of the proposed method.
6-	In result tables, please sort methods according to their overall performance, and correct bolding in table 2 (protonet 5-way-5 shot MetaNorm is not best performing method) and highlight setting where methods have very similar performance (as in Table 14) for clarity.
7-	Provide a clear list of contributions at the end of the introduction section
8-	While KL divergence and hypernetworks are well known terms, it would make the paper more accessible to add a sentence (and equation in the case of the KL divergence) or two describing the terms. In particular, hypernetworks are generally used to characterise weight generators for entire architectures and might lead to confusion.
9-	It could be nice to provide more attention to how this work relates to conditional batch norm works, and whether they can be complementary.

---

> ### Author Response · Authors · 2020-11-23
> **Response to AnonReviewer4**
>
> We thank *AnonReviewer4* for constructive comments and clear guidance.
>
> **Regarding presentation, too many implementation details and descriptions of the experiments are missing from the main paper (and in certain cases, missing altogether). Regarding datasets and implementation, it is ok to provide non essential details in the appendix (especially considering the large number of datasets considered). However, no information is provided at all in the main paper, which makes it very difficult to understand the setting of the experiments.**
>
> We regret our experimental description has confused you, and hope it is still mendable with the extra page. We have expanded the experimental details in the main paper and in the appendix, which we detail per item below.
>
> **For example, in Table 1 and Figure 2, it is not known on which datasets experiments are run, and it is never mentioned in the main text (for table 1) what FSL method is employed.**
>
> In Table 1 and the subsection on *Effect of KL Term* we clarify our method runs on miniImageNet for few-shot learning and PACS for domain generalization. In Figure 2 we report on the same two sets. We have updated the caption and the subsection on *Impact of Target Set Size*. Thank you.
>
> **In addition, Table 2 experiments comparing MetaNorm to different approaches sorely lacks description. The approach is compared to 9 different algorithms, none of which are given a description or reference to learn more about the method.**
>
> The reviewer is right, we apologize. We have added the reference of each algorithm. In the subsection on *Sensitivity to Domains* we stated the baseline as "The baseline normalization uses the statistics from the source domains for the batch normalization of the target domain".
>
> **Finally, experiments in Figure 2 would be a lot more interesting if compared to standard methods. It would be interesting to see how the proposed strategy allows to be more sample efficient and reach stronger performance than transductive batch norm in situations where sample size is the smallest. As this is one of the cited main limitations of TBN, this experiment is highly important.**
>
> We have added the TBN by [Bronskill et al. (2020)](https://arxiv.org/pdf/2003.03284.pdf) in Figure 2 and add to the discussion in  *Impact of Target Set Size*: "The experimental results show that TBN has limited effect on the target size both on the 5-way, 1-shot and 5 way, 5-shot tasks". To avoid clutter, we only show the results of TBN with VERSA. The same trend holds also for MAML and ProtoNets.
>
> **With regards to related work, I would suggest to move the section after the introduction, where it provides much better context to facilitate method comprehension, in particular regarding the description of the TBN strategy.**
>
>  We moved the related work section after the introduction.
>
> **Authors should also comment on how their work, and in particular FSL domain generalisation setting, relates to cross domain few-shot learning works, e.g. Tseng et al ICLR 2020, Cross-Domain Few-Shot Classification via Learned Feature-Wise Transformation It is currently presented as a completely new approach to FSL, and appears to ignore past cross domain FSL works. Please provide additional context regarding such works.**
>
> We add in Section  *MetaNorm for Few-Shot Domain Generalization*: "Others have considered the related task of cross-domain few-shot learning, e.g. [Tseng et al. (2020)](https://arxiv.org/abs/2001.08735) and [Guo et al. (2020)](https://arxiv.org/abs/1912.07200). Different from their settings, our few-shot domain generalization is more challenging as the support and query set are from *different* domains in the meta-test stage and the target domain is also unseen throughout the training stage".
>
> **RECOMMENDATION**
>
> 1. We have reorganized the section *Experiments Results*.
>
> 2. We moved the related work section after the introduction.
>
> 3. We have provided a discussion about cross-domain few-shot learning in the section of *related work*.
>
> 4. We have provided the suggested comparison to standard TBN in Figure 2.
>
> **Additional suggestions**
>
> 5. We have added an overview figure of the proposed method in Figure 3 of the appendix.
>
> 6. There is no consistent way to sort the methods because they are implemented under three different models. We have corrected the bolding in Table 2 and Table 14.
>
> 7. We have provided a list of contributions at the end of the introduction.
>
> 8. We have added some sentences about KL divergence and hypernetworks in our methodology.
>
> 9. We have updated the related work section to include a discussion on conditional batch normalization.

---

### Author Response · Authors · 2020-11-23
**Summary of Changes**

**We thank all AnonReviewers for their insightful reviews, sharp comments and supportive suggestions. Here, we provide a summary of the updates made in the new version, as suggested by the reviewers.**

## Main manuscript
The following updates have been incorporated by using the extra page for the main manuscript:
* We have provided an explicit list of contributions at the end of the introduction.
* We have expanded related work, by adding references and discussion related to *batch normalization for domain adaptation and domain generalization*, and a new subsection about *conditional batch normalization*. We also move the related work after the introduction.
* We have added our intuition why MetaNorm generates proper statistics for new domains, and we provide a new comparison for domain generalization using ground truth statistics in the subsections *MetaNorm for Domain Generalization*, *Sensitivity to Domains*, and Table 4.
* We have expanded the experiment with varying query size in Figure 2 with standard TBN.
* We have added a detailed description and reference of each algorithm in *Experimental Results*.
 * We have clarified what experimental results in Table 2 are provided by [Bronskill et al. (2020)](https://arxiv.org/pdf/2003.03284.pdf), and which ones are computed by us.
* We have added a discussion about cross-domain few-shot learning in *MetaNorm for Few-Shot Domain Generalization*.
* We have clarified in the *Methodology*, the KL divergence and hypernetworks, the total loss,  the description of $m$ in equation 3, and the parameters of the KL expressions in equations 4, 8, 10.

## Appendix
The following updates have been inserted in the Appendix:
* We have provided an overview figure of the implementation of our MetaNorm for few-shot learning in Figure 3.
* We have added the details and codebases about the experimental setting for MAML, ProtoNets, and VERSA in Appendix D.
* We have added the detailed descriptions about the structure of $f_{\mu}^l(\cdot)$ and $f_{\mu}^l(\cdot)$ in Table 9, 10.
 * We have clarified what experimental results in Tables 14, 15, and 16 are provided by [Bronskill et al. (2020)](https://arxiv.org/pdf/2003.03284.pdf), and which ones are computed by us.
*  We have added the convergence analysis and plot the training loss versus iterations for the ProtoNets algorithm in Appendix H and Figure 4.
* We have implemented experiments on the effect of different numbers of units in the hidden layer on performance and added this experiment in Table 17.

---

### Decision · Program_Chairs · 2021-01-07
**Final Decision**

**Decision:**

Accept (Poster)

**Comment:**

This paper proposes an lightweight method for cross-domain few-shot learning, using a meta-learning approach to predict batch normalization statistics.
After the extensive paper revisions and discussion, the reviewers all agreed that this paper is above the bar for acceptance, assuming that the authors will include results for both the standard and expanded target set size in the final version of the paper. The authors are strongly encouraged to include these results in the camera-ready version of the paper.